# Defending Membership Inference Attacks via Privacy-aware Sparsity Tuning

## Abstract

Over-parameterized models are often vulnerable to membership inference attacks, which aim to determine whether a specific sample is included in the training set of a target. Previous weight regularization approaches typically impose uniform penalties on all parameters, leading to a suboptimal trade-off between model utility and privacy. In this work, we first show that only a small fraction of the parameters substantially impact privacy risk. Motivated by this analysis, we propose Privacy-aware Sparsity Tuning (**PAST**)—a novel privacy-preserving training method—by employing adaptive penalties to different model parameters. Our key idea behind PAST is to promote sparsity in model parameters that significantly contribute to privacy leakage risk. In particular, we construct the adaptive weight for each parameter based on its privacy sensitivity, i.e., the gradient of the loss gap with respect to the parameter. By using PAST, the network reduces the loss gap between members and non-members, thereby improving resistance to privacy attacks while preserving model utility. Extensive experiments consistently demonstrate the superiority of our defense method across various datasets, different network architectures, and diverse attack methods, achieving superior performance in the privacy-utility trade-off.

## 1 Introduction

Modern deep neural networks are typically highly parameterized, where the number of model parameters greatly exceeds the size of the training set (Zhang et al., 2021). While the huge amount of parameters empowers the models to achieve impressive performance across various tasks, the strong capacity also makes them vulnerable to membership inference attacks (MIAs) (Shokri et al., 2017). In MIAs, attackers aim to infer whether a particular sample is included in the target model's training set. Membership inference can raise significant security and privacy concerns when the target models are trained on sensitive data, such as health care (Paul et al., 2021), financial services (Mahalle et al., 2018), and DNA sequence analysis (Arshad et al., 2021). Therefore, it is crucial to develop privacy-preserving training methods on the model end to defend against membership inference attacks.

The main challenge in defending against MIAs stems from the large number of model parameters, which can inadvertently disclose information about the training data (Tan et al., 2022). Therefore, previous work sought to reduce the over-complexity of neural networks by weight regularization, such as $\ell_1$ or $\ell_2$ regularization (Hamida et al., 2023). These regularization techniques impose uniform penalties on all parameters with large values, thereby reducing overfitting to the training data and encouraging neuron weights to shrink towards zero. However, if model parameters contribute differently to sensitive information leakage, uniform penalties may lead to a suboptimal trade-off between model utility and privacy. This naturally motivates us to raise a reflective question: *Do all model parameters contribute equally to privacy leakage?*

In this work, we first perform an empirical analysis of the privacy sensitivity[1] of model parameters, revealing that different parameters contribute differently to privacy leakage. In particular, we take the loss gap between member and non-member examples as a proxy for privacy risk and compute its gradient with respect to each

---

[1]In this work, *privacy sensitivity* refers to the gradient of the loss gap between members and non-members with respect to each model parameter, distinguishing it from the existing notion of sensitivity in privacy literature.

model parameter. We find that only a small fraction of parameters substantially impact the privacy risk, whereas the majority have a negligible effect. Consequently, applying uniform penalties to all parameters is either inefficient or suboptimal for defending against MIAs, potentially leading to unnecessary restrictions on the model's capacity and degrading its utility.

To address this issue, we propose Privacy-Aware Sparsity Tuning (PAST), an adaptive privacy-preserving regularization method that dynamically adjusts penalty strength for different model parameters in a deep neural network. The key idea behind PAST is to promote sparsity in model parameters that significantly contribute to privacy leakage risk. In particular, we assign parameter-specific regularization strengths to each model parameter based on its privacy sensitivity, quantified by the gradient of the loss gap with respect to the parameter. Our method imposes strong regularization only on a small subset of privacy-sensitive parameters, thereby preserving model utility. Trained with the proposed adaptive privacy-aware regularization technique, the model shrinks the loss gap between members and non-members, leading to strong resistance to membership inference attacks.

To validate the effectiveness of our method, we conduct experiments across a variety of datasets and diverse neural network architectures. Specifically, we conduct extensive evaluations on both tabular (Texas100 (Texas Department of State Health Services, 2006), Purchase100 (DMDave et al., 2014)) and image datasets (CIFAR-10/100 (Krizhevsky et al., 2009), ImageNet (Russakovsky et al., 2015)) using various model architectures, including ResNet (He et al., 2016), DenseNet (Huang et al., 2017), SqueezeNet (Iandola et al., 2016) and MobileViT (Mehta & Rastegari, 2021). The results demonstrate that our PAST achieves superior privacy-utility trade-offs across a variety of attacks, including the state-of-the-art RMIA attack (Zarifzadeh et al., 2024), compared to other defense methods. For example, compared with the LabelSmoothing defense, our PAST method significantly reduces the attack advantage of loss-based attack from 51.2% to 33.8%, while maintaining a comparable test accuracy. In addition, our method is implemented during the fine-tuning phase, making it directly applicable to improve other defense methods. Our contributions are summarized as follows:

- We first investigate the importance of model parameters in privacy risk. Our empirical results indicate that only a few parameters have a substantial impact on the privacy risk, while the majority have a negligible effect. This motivates an MIA defense method that focuses on a small number of privacy-sensitive parameters.

- We introduce PAST – a novel and effective defense method, which promotes sparsity in parameters that significantly contribute to privacy leakage. We show that PAST can effectively achieve superior privacy-utility trade-offs across a variety of attacks.

- We conduct extensive experiments on various datasets under diverse network architectures, demonstrating that our PAST achieves superior performance to existing defenses. Moreover, our method is applicable to improve other defense methods.

## 2 Preliminaries

**Setup**    In this paper, we study membership inference attacks on $K$-class classification tasks. Let the feature space be $\mathcal{X} \subset \mathbb{R}^d$ and the label space be $\mathcal{Y} = \{1, \ldots, K\}$. Let $(\boldsymbol{x}, y) \in (\mathcal{X} \times \mathcal{Y})$ denote an example containing an instance $\boldsymbol{x}$ and a real-valued label $y$. Given a training dataset $\mathcal{S} = \{(\boldsymbol{x_i}, y_i)\}_{i=1}^N$ *i.i.d.* sampled from the data distribution $\mathcal{P}$, our goal is to learn a model $h_\theta$ with trainable parameters $\theta \in \mathbb{R}^p$, that minimizes the following expected risk:

$$\mathcal{R}(h_\theta) = \mathbb{E}_{(\boldsymbol{x}, y) \sim \mathcal{P}}[\mathcal{L}(h_\theta(\boldsymbol{x}), y)], \tag{1}$$

where $\mathbb{E}_{(\boldsymbol{x}, y) \sim \mathcal{P}}$ denotes the expectation over the data distribution $\mathcal{P}$ and $\mathcal{L}$ is a loss function (e.g., cross-entropy loss) for classification. In modern deep learning, the neural network $h_\theta$ is typically overparameterized, allowing it to easily disclose training data information (Tan et al., 2022).

**Membership inference attacks**  Given a data point $(\boldsymbol{x}, y)$ and a trained target model $h_{\mathcal{S}}$, the goal of membership inference attacks is to identify whether a sample is included in the training set $\mathcal{S}$ of the target model (Shokri et al., 2017; Yeom et al., 2018; Salem et al., 2019a). In MIAs, it is generally assumed that attackers can query the model's predictions $h_\theta(\boldsymbol{x})$ and access the model's parameters. Here, we consider the standard setting for MIAs (Irolla & Châtel, 2019), in which the attacker has access to model architecture and the training data distribution $\mathcal{P}$.

In the process of attack, the attacker has access to the query set $Q = \{(\boldsymbol{z}_i, m_i)\}_{i=1}^{J}$, where $\boldsymbol{z}_i$ denotes the $i$-th data point $(\boldsymbol{x}_i, y_i)$, and $m_i$ denotes whether the given data point $(\boldsymbol{x}_i, y_i)$ is a member of the training dataset $\mathcal{S}$, i.e., $m_i = \mathbb{I}[(\boldsymbol{x}_i, y_i) \in \mathcal{S}]$. In particular, the query set $Q$ contains both member and non-member samples, drawn from the data distribution $\mathcal{P}$. Then, the attacker $\mathcal{A}$ can be formulated as a binary classifier, which predicts $m_i \in \{0, 1\}$ for a given example $(\boldsymbol{x}_i, y_i)$ and a target model $h_\theta$: $\mathcal{A}(\boldsymbol{x}_i, y_i; h_\theta) \to \{0, 1\}$.

**Weight regularization**  The privacy risk of deep neural networks is often associated with their over-parameterized nature. Intuitively, the large number of parameters enables the model to capture extensive information about the training data, potentially leading to unintended privacy leaks. Previous work theoretically shows that increasing the number of model parameters renders them more vulnerable to membership inference attacks (Tan et al., 2022). To address this issue, previous work employed weight regularization techniques, such as $\ell_1$ and $\ell_2$ regularization, to mitigate membership inference attacks. (Hoerl & Kennard, 1970; Tibshirani, 1996; Schmidt et al., 2007). Formally, the weight regularization can be formalized as:

$$\mathcal{R}_{\text{reg}}(h_\theta) = \mathbb{E}_{(\boldsymbol{x}, y) \sim \mathcal{P}}[\mathcal{L}(h_\theta(\boldsymbol{x}), y)] + \lambda R(h_\theta), \tag{2}$$

where $\mathcal{L}(\cdot)$ denotes the classification loss, $\lambda$ is the hyperparameter that controls the strength of the regularization term, and $R(\cdot)$ is typically designed to penalize the complexity of the model parameters $h_\theta$. For example, regularizations of $\ell_1$ and $\ell_2$ are used to penalize the norm of model parameters as follows: $R(h_\theta) = \|\theta\|_r^r$, where $r$ denotes the order of the norm.

Previous work (Tan et al., 2022) shows that reducing the number of effective model parameters can mitigate vulnerability to MIAs, for example, by leveraging the sparsification effect of $\ell_1$ regularization. However, this comes at the cost of inferior generalization performance (utility) due to the "double descent" effect (Belkin et al., 2019; 2020; Dar et al., 2021), wherein generalization error can decrease as the degree of model over-parameterization increases. More importantly, existing regularization methods apply uniform penalties to all parameters, ignoring their potentially varying importance in terms of privacy leakage.

## 3   Method: Privacy-Aware Sparsity Tuning

In this section, we start by analyzing the privacy sensitivity of model parameters and find that most parameters contribute only marginally to the privacy leakage. Subsequently, we design an adaptive privacy-aware regularization that accounts for the privacy sensitivity of each parameter.

### 3.1   Motivation

In this part, we aim to figure out whether the model parameters are equally important in terms of privacy risk. In particular, we perform standard training with ResNet-18 (He et al., 2016) on CIFAR-10 (Krizhevsky et al., 2009), using SGD with a momentum of 0.9, a weight decay of 0.0005, and a batch size of 128. We set the initial learning rate to 0.01 and decrease it using a cosine scheduler (Loshchilov & Hutter, 2017) throughout the entire training process. In the experiment, we construct the member set $\mathcal{S}_m$ and the non-member set $\mathcal{S}_n$ by randomly sampling 10,000 examples from the training set and the test set, respectively.

**Loss gaps as a proxy for privacy risk**  In this study, we use the empirical loss gap between member and non-member examples as a proxy for privacy leakage risk. Formally, we define loss gap as:

$$\mathcal{G}(\mathcal{S}_m, \mathcal{S}_n; h_\theta) = |\bar{\mathcal{L}}(\mathcal{S}_m; h_\theta) - \bar{\mathcal{L}}(\mathcal{S}_n; h_\theta)| \tag{3}$$

$$= |\frac{1}{|\mathcal{S}_m|} \sum_{(\boldsymbol{x}, y) \in \mathcal{S}_m} \mathcal{L}(h_\theta(\boldsymbol{x}), y) - \frac{1}{|\mathcal{S}_n|} \sum_{(\boldsymbol{x}, y) \in \mathcal{S}_n} \mathcal{L}(h_\theta(\boldsymbol{x}), y)|. \tag{4}$$

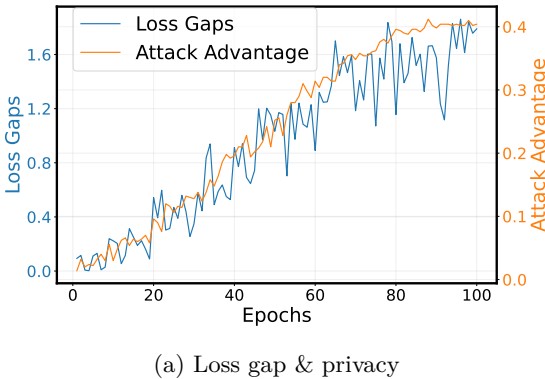 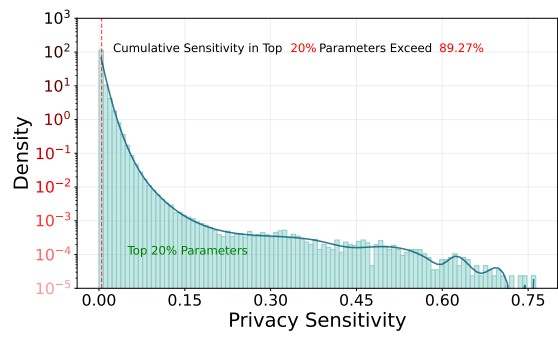

(a) Loss gap & privacy          (b) Privacy sensitivity distribution

Figure 1: (a) Loss gaps and attack advantage during training on CIFAR-10 with ResNet-18. The attack advantage increases as the loss gap increases during the training. (b) The privacy sensitivity distribution across model parameters. The cumulative sensitivity in the top 20% parameters accounts for more than 89.27% of the total, indicating only a small fraction of parameters substantially impacts the privacy risk.

Specifically, we compute the gap between the average losses of member and non-member examples. A larger loss gap indicates a higher risk of privacy leakage, suggesting that the model is more vulnerable to membership inference attacks. Prior work shows that the loss function can sufficiently determine the optimal attacks in membership inference (Sablayrolles et al., 2019a). Empirically, as shown in Figure 1a, we observe that the attack advantage (See the definition in Section 4.1) increases as the loss gap increases during the training, supporting the use of the loss gap as a reliable proxy for privacy leakage risk. Furthermore, in Appendix B, we provide detailed theoretical and empirical justification for using the loss gap as a privacy proxy and demonstrate its rationale and generalizability across various attack methods.

**Most model parameters contribute only marginally to the privacy leakage risk**   To investigate the impact of individual model parameters on privacy leakage, we compute each parameter's privacy sensitivity and examine how much each parameter contributes to the overall leakage risk. In particular, we denote the privacy sensitivity of a parameter $\theta_i$ by measuring the gradient of the loss gap with respect to the parameter. Figure 1b illustrates the privacy sensitivity distribution of model parameters. The results show that only a small fraction of parameters substantially impact the privacy risk, whereas the majority have little effect. For example, 97% of the model parameters have privacy sensitivities lower than 0.1. In addition, the cumulative sensitivity in the top 20% parameters exceeds 89.27% of the total privacy sensitivity. Our empirical observations suggest that different parameters make highly unequal contributions to privacy leakage risk. Therefore, applying uniform penalties to all parameters is inefficient in defending against MIAs and may unnecessarily restrict the model's capacity. In Appendix D, we further present privacy sensitivity distributions using other metrics as proxies for privacy leakage risk, such as entropy and M-entropy, which yield similar outcomes. In light of the findings, we consider applying adaptive weight regularization, which focuses on the most privacy-sensitive model parameters rather than the entire parameter set.

### 3.2   Method

Our previous analysis shows that most model parameters contribute only marginally to privacy leakage risk, suggesting that a well-designed defense should focus on the most privacy sensitive parameters rather than uniformly penalizing all parameters. Thus, our core idea is to promote sparsity specifically among the model parameters that contribute most to privacy leakage risk.

**Privacy-Aware Sparsity Tuning**   In this work, we introduce Privacy-Aware Sparsity Tuning (**PAST**), an adaptive regularization that employs different penalty strengths for each parameter in a deep neural network. Formally, the objective function of PAST is given by:

$$\mathcal{R}_{\text{PAST}}(h_\theta) = \mathbb{E}_{(\boldsymbol{x},y)\sim\mathcal{P}}[\mathcal{L}(h_\theta(\boldsymbol{x}),y)] + \lambda R(h_\theta)$$
$$= \mathbb{E}_{(\boldsymbol{x},y)\sim\mathcal{P}}[\mathcal{L}(h_\theta(\boldsymbol{x}),y)] + \lambda \sum_i \gamma_i |\theta_i|, \tag{5}$$

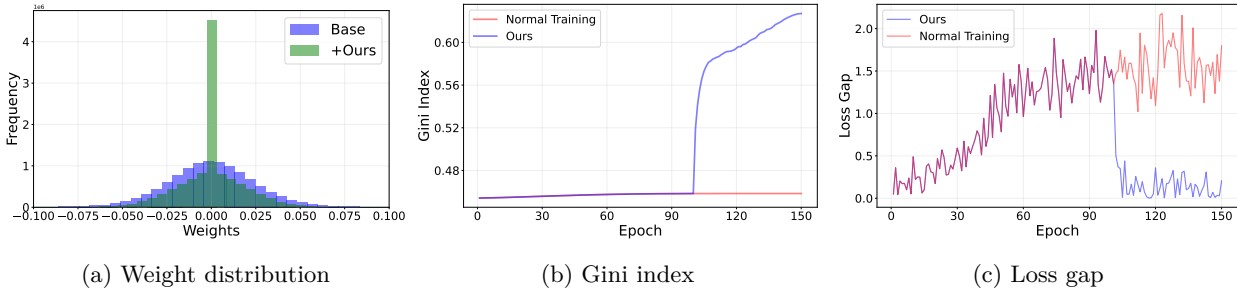

(a) Weight distribution           (b) Gini index           (c) Loss gap

Figure 2: (a) Comparison of weight distribution before (Base) and after applying PAST (+Ours) on CIFAR-10. The weights of PAST are more concentrated around 0 and are thus sparser than the baseline. (b) Comparison of the Gini Index before (Base) and after applying PAST. The Gini index (a sparsity metric defined in Appendix E) quickly increases after applying PAST, which demonstrates the sparsity effect of PAST. (c) The impact of PAST on the loss gap. The loss gap quickly decreases after applying PAST, thereby enhancing resistance to privacy attacks.

where $\lambda$ is the hyperparameter controlling the regularization term's importance and $\gamma_i$ denotes the adaptive weight of the parameter $\theta_i$. We expect larger weights for parameters with higher privacy sensitivity, and smaller weights for those with lower sensitivity. Using the $\ell_1$ norm, the regularization can encourage sensitive parameters to be zero, thereby improving defense against MIAs.

In particular, we modulate the intensity of regularization for each model parameter based on its privacy sensitivity, i.e., the gradient of the loss gap with respect to the parameters. Let $\mathcal{S}_m$ and $\mathcal{S}_n$ denote the subsets of members and non-members, respectively. For brevity, we use $\mathcal{G}_\theta$ to denote the loss gap $\mathcal{G}(\mathcal{S}_m, \mathcal{S}_n; h_\theta)$ of the model $h_\theta$ on $\mathcal{S}_m$ and $\mathcal{S}_n$. We then compute the privacy sensitivity of each parameter $\theta_i$ and normalize it within the corresponding module (e.g., a linear layer). Formally, we define the adaptive weight term as:

$$\gamma_i = \frac{|\theta^{(\ell)}| \ |\nabla_{\theta_i} \mathcal{G}_\theta|}{\sum_{\theta_j \in \theta^{(\ell)}} |\nabla_{\theta_j} \mathcal{G}_\theta|}, \tag{6}$$

where $|\theta^{(\ell)}|$ denotes the number of parameters in the $\ell$-th module, and $\nabla_{\theta_i} \mathcal{G}_\theta$ denotes the gradient of the loss gap with respect to parameter $\theta_i$. Notably, the module-wise normalization eliminates the influence of different gradient scales across modules, ensuring that $\gamma_i$ reflects the relative privacy sensitivity of each parameter within its module.

In practice, we introduce a hyperparameter $\alpha$ to modulate the parameter-wise regularization weights $\gamma_i$. Concretely, the final regularization term of PAST can be formulated as:

$$R(h_\theta) = \sum_i \gamma_i^\alpha |\theta_i|, \tag{7}$$

when $\alpha = 0$, the regularization is equivalent to the standard $\ell_1$ regularization. As $\alpha$ increases, the regularization increasingly focuses on the small subset of parameters with high privacy sensitivity. This allows users to flexibly control the privacy-utility trade-off by adjusting $\alpha$ based on their desired level of privacy protection. The adaptive parameter-wise weighting scheme reduces penalties on privacy-insensitive parameters while imposing stronger penalties on privacy-sensitive parameters. By incorporating PAST into training, the model can achieve a superior trade-off between privacy and utility.

**Empirical Analysis of PAST** Our PAST not only stringently regularizes sensitive parameters but also preserves model utility by sparing less sensitive parameters from excessive regularization. Specifically, to evaluate the sparsity effect of our PAST, we conduct experiments on CIFAR-10 using 100 epochs of standard training, followed by 50 epochs of sparse fine-tuning with our PAST. As shown in Figures 2a and 2b, the results show our method mitigates over-parameterization both intuitively and quantitatively: (1) more parameters are concentrated around 0 in the weight distribution after PAST tuning; (2) the Gini index, a sparsity metric defined in previous work (Hurley & Rickard, 2009; Goswami et al., 2021) and presented in Appendix E, also increases significantly. We also present the variation in loss gap before and after

PAST tuning in Figure 2c. The results show that the loss gap sharply decreases after applying our PAST, demonstrating our method's capability to reduce the loss gap and enhance resistance to privacy attacks.

**Fine-tuning with PAST**  Standard weight regularization is usually employed from the beginning of model training to alleviate overfitting. This makes it challenging to achieve an optimal trade-off between privacy and utility, as uniform regularization restricts the model's capacity. Instead, we introduce PAST after the model has converged in terms of the training loss. Specifically, we employ our PAST to fine-tune the models after standard training. The tuning mechanism of PAST enables our method to be applied to trained models without retraining from scratch. As a result, existing models can be fine-tuned with PAST to mitigate the risk of privacy leakage, which enhances the practical applicability of the method. In addition, our method is plug-and-play, making it directly applicable to existing defense methods. Figure 5b in Section 4.2 shows that our method can improve the performance of existing defenses.

## 4 Experiments

### 4.1 Setup

**Datasets and models**  In our evaluation, we conduct experiments on various types of datasets, including two tabular datasets (e.g., Texas100 (Texas Department of State Health Services, 2006) and Purchase100 (DMDave et al., 2014)) and three image datasets (e.g., CIFAR-10, CIFAR-100 (Krizhevsky et al., 2009), and ImageNet (Russakovsky et al., 2015)). Following previous work (Liu et al., 2024b), we randomly split each dataset into six subsets, which are used as the training, testing, and inference set for target and shadow models. For instance, the CIFAR-10 dataset is randomly partitioned into six equal subsets, with each subset comprising 10,000 samples. Our method use the inference set as non-members during PAST fine-tuning. Additionally, the inference set is utilized by adversarial training algorithms that incorporate adversary loss, such as Mixup+MMD (Li et al., 2021) and adversarial regularization (Nasr et al., 2018). To evaluate the performance of our method across various model architectures, we conduct experiments on different model architectures, including DenseNet121 (Huang et al., 2017), ResNet18 (He et al., 2016), SqueezeNet (Iandola et al., 2016), and MobileViT (Mehta & Rastegari, 2021). Additionally, following prior work (Nasr et al., 2018; Jia et al., 2019), we conduct experiments on multilayer perceptron (MLP) models with tabular datasets.

**Experimental details**  We train the models using SGD with a momentum of 0.9, a weight decay of 0.0005, and a batch size of 128. We set the initial learning rate to 0.01 and drop it using a cosine scheduler (Loshchilov & Hutter, 2017) with $T_{max} = epochs$. For CIFAR-10, we conduct training using an 18-layer ResNet (He et al., 2016), with 100 epochs of standard training and 50 epochs of PAST tuning. In the case of ImageNet and CIFAR-100, we employ a 121-layer DenseNet (Huang et al., 2017) with 100 epochs of standard training and 20 epochs of PAST tuning. For Texas100 and Purchase100, we follow previous work (Nasr et al., 2018; Jia et al., 2019) and train multilayer perceptrons (MLPs) for 100 epochs using standard training, followed by 20 epochs of PAST fine-tuning. In the main experiments, we follow the hyperparameter tuning protocols established by previous work (Chen et al., 2022). For the hyperparameter regularization strength $\lambda$ in Equation (5), we set the scale factor to 1e-3 for CIFAR-10, 1e-4 for CIFAR-100, and 1e-5 for all other datasets. For the hyperparameter $\alpha$ defined in Equation (7), we employ grid search to achieve better performance. The detailed parameter settings are provided in Appendix F. In practice, we sweep $\alpha$ over the specified range, recording the task accuracy and corresponding attack advantage at each value to plot the privacy-utility trade-off curve. In practice, we apply this procedure to each model and dataset independently.

**Attack methods**  In our work, we conduct experiments with ten membership inference attack methods spanning four categories: (1) Neural Network-based Attack (NN) (Shokri et al., 2017; Hu et al., 2022), leverages full logits prediction as input for attacking the neural network model. (2) Metric-based Attack, employs specific metric computations followed by comparison with a preset threshold to ascertain a data record's membership. The metrics we chose for our experiments include Correctness, Loss (Yeom et al., 2018), Confidence, Entropy (Salem et al., 2019a), and Modified-Entropy (M-entropy) (Song & Mittal, 2021). (3) Augmentation-based Attack (Choquette-Choo et al., 2021), utilizes prediction data derived through data

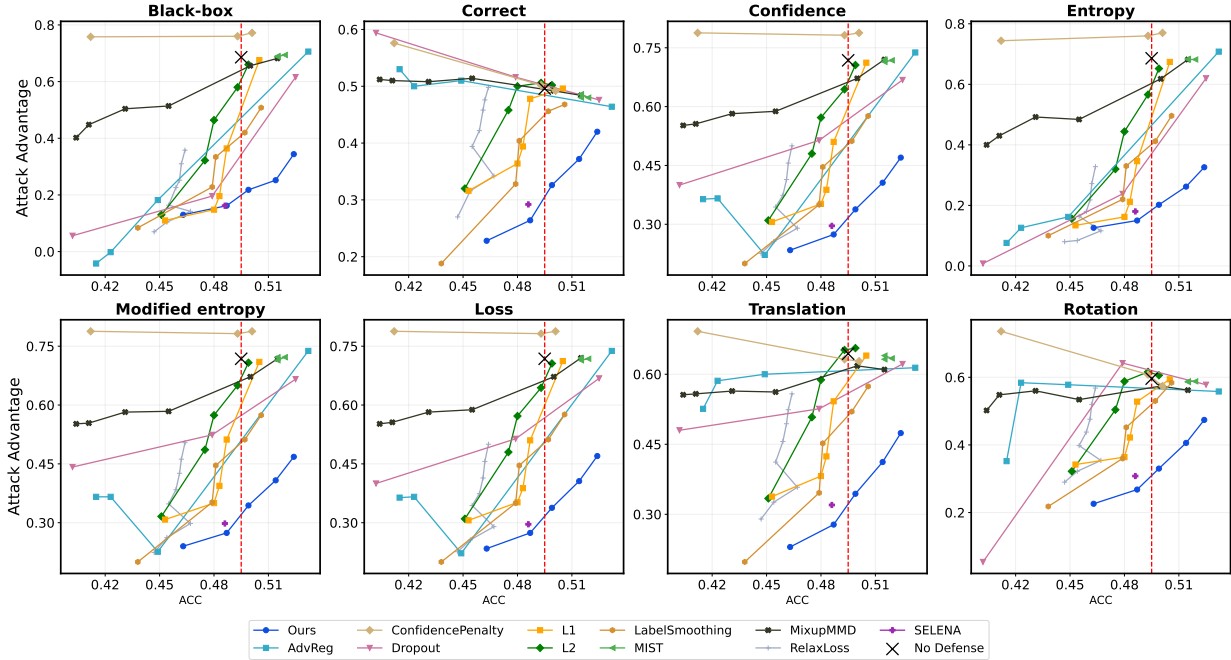

Figure 3: Comparisons of various defense mechanisms on CIFAR-100 dataset utilizing Densenet121 architecture. Each subplot is allocated to a distinct attack method, wherein individual curves represent the performance of a defense mechanism under different hyperparameter settings. The horizontal axis represents the target models' test accuracy, with higher values indicating better performance. The vertical axis represents the corresponding attack advantage, where lower values indicate better defense performance. To underscore the disparity between the defense methods and the vanilla (undefended model), we plot the dotted line originating from the vanilla model's results.

augmentation techniques as inputs for a binary classifier model. In this method, we specifically implemented rotation and translation augmentations. (4) Inference-based Attack, including LiRA (Carlini et al., 2022), and RMIA (Zarifzadeh et al., 2024), trains multiple inference models on different training sets and utilizes signals from these models to estimate distributions of members and non-members. During inference, this method predicts whether a given sample belongs to the member or non-member distribution, based on statistical inference. For all inference-based attacks, we train 32 inference models. For each instance, we assign 16 models to the in-model set and the remaining 16 to the out-model set.

**Defense baselines** We compare PAST with ten defense methods: MIST (Li et al., 2024), SELENA (Tang et al., 2022), RelaxLoss (Chen et al., 2022), Mixup+MMD (Li et al., 2021), Adversarial Regularization (AdvReg) (Nasr et al., 2018), Dropout (Srivastava et al., 2014), Label Smoothing (Guo et al., 2017), Confidence Penalty (Pereyra et al., 2017), $\ell 1$ and $\ell 2$ regularization (Shokri et al., 2017).

**Evaluation metrics** To comprehensively assess the impact of our method on privacy and utility, we employ three evaluation metrics that encapsulate utility, privacy, and the trade-off between the two. Utility is gauged by test accuracy of the target model. Privacy is measured by attack advantage (Yeom et al., 2018):

$$Adv(\mathcal{A}) = 2\Pr(\mathcal{A}(h_{\mathcal{S}}(\boldsymbol{x}), y) = m) - 1, \tag{8}$$

where the notations are defined in Section 2. To assess the trade-offs between utility and privacy, we utilize the $P_1$ score (Paul et al., 2021). The score is defined as:

$$P_1 = 2 \times \frac{\text{Acc} \times (1 - \text{Adv})}{\text{Acc} + (1 - \text{Adv})}, \tag{9}$$

where $Acc$ denotes test accuracy, and $Adv$ denotes attack advantage on the target model. In addition, we follow the settings from prior work (Zarifzadeh et al., 2024) to evaluate the performance of inference-based

attacks. Specifically, we evaluate the attack performance by measuring the following metrics: (1) AUC, the area under the receiver operating characteristic curve; (2) the true positive rate (TPR) when the false positive rate (FPR) is 0.1% (TPR@0.1%FPR).

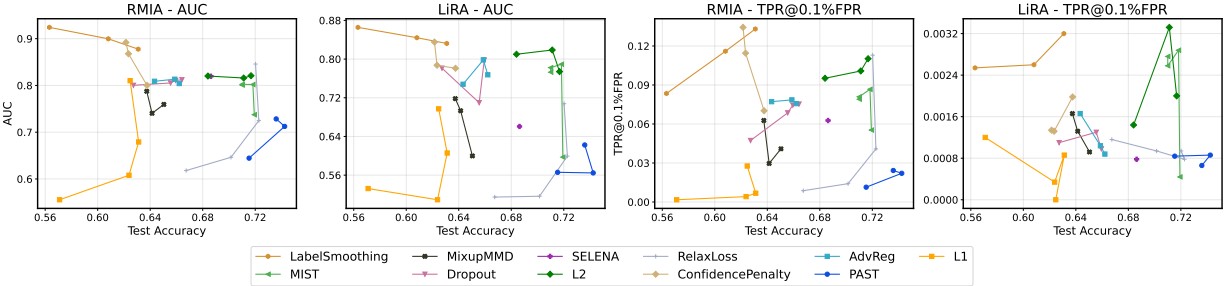

Figure 4: AUC and TPR@0.1%FPR vs. Test Accuracy for the state-of-the-art RMIA and LiRA on CIFAR-10 with different defenses. The x-axis denotes test accuracy, while the y-axis denotes attack performance, with lower values on the y-axis indicating better defense performance. Our PAST achieves a superior trade-off compared to other defense methods.

## 4.2 Results

**Can PAST achieve better performance than existing defense methods?** In Figure 3 and Figure 4, we plot privacy-utility curves to show the privacy-utility trade-offs. The results show that the privacy–utility curves of our method consistently lie below those of other defense methods, demonstrating that PAST achieves a superior privacy-utility trade-off compared with existing defenses. For example, compared with the LabelSmoothing defense, our PAST method significantly reduces the attack advantage of loss-based attack from 51.2% to 33.8%, while maintaining a comparable test accuracy (49.90% for PAST vs. 49.74% for LabelSmoothing). Moreover, we evaluate the performance of our method and baselines using state-of-the-art attack methods (e.g., LiRA and RMIA). The results in Figure 4 further demonstrate that our method outperforms other defense methods in terms of both AUC and TPR@0.1%FPR. In Appendix H, we further provide the privacy–utility curves of our method and the baseline on CIFAR-10, demonstrating that our method consistently outperforms other defenses. In addition, we conduct experiments to evaluate the performance of our PAST on diverse model architectures, including DenseNet121, ResNet18, SqueezeNet, and MobileViT. The results in Appendix I show that our PAST achieves superior performance across different model architectures. In summary, our results show that PAST consistently outperforms existing defense methods across comprehensive attack methods, various evaluation metrics, and different model architectures. Additionally, we report statistical metrics of our method on CIFAR-10 with ResNet18 across different seeds in Appendix J. The experimental results show that the small standard deviation of the P1 score indicates that the improvements achieved by PAST are stable across different random seeds, rather than a result of random chance.

**Is our PAST effective across different datasets?** To evaluate the effectiveness of our proposed method across diverse datasets, we conduct experiments on two tabular datasets (e.g., Texas100, Purchase100) and three image datasets (e.g., CIFAR-10, CIFAR-100, and ImageNet). In the experiment, we use a fixed value of $\alpha = 2.5$ across all datasets, without any dataset-specific hyperparameter tuning. Details of hyperparameters and model architectures are reported in Section 4.1. Specifically, we compute the P1 score based on the average attack advantage and accuracy across different attack methods. In Figure 5a, the results show that our method yields a higher P1 score after applying PAST, demonstrating the effectiveness of our method on different datasets.

**Can our PAST improve other defense methods?** As described in Section 3.2, our method is applied during the fine-tuning phase. Thus, our method is different from other defense methods that are incorporated from the beginning of model training. In practice, our method is compatible with existing defense methods

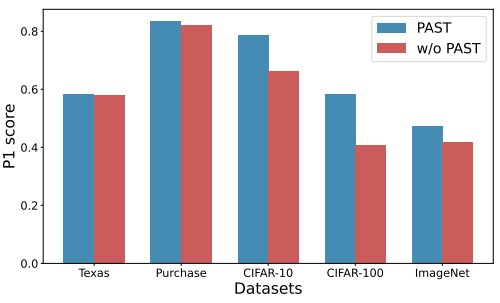
(a) Performance on various datasets

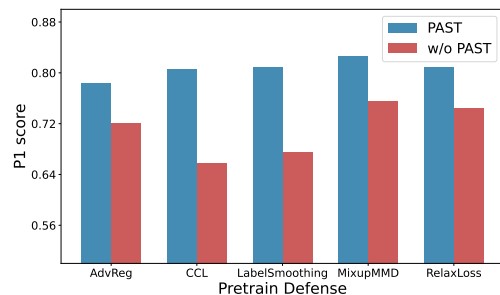
(b) Performance against various defenses

Figure 5: Impact of PAST on P1 score on different datasets and defense methods. The definition of the P1 score is presented in Equation (9), where a large value is preferred. *PAST* indicates the results after applying our PAST, and *w/o PAST* denotes the results obtained without PAST. Our method is consistently effective on various datasets and improves the performance of current defense methods.

and can further improve their performance by fine-tuning the defended models with PAST. To evaluate the effectiveness of our PAST in improving other defense methods, we conduct experiments on CIFAR-10 using ResNet18 by further fine-tuning the model for 50 epochs with $\alpha = 1.5$. In our experiment, we compare our PAST with various defense methods, including LabelSmoothing, AdvReg, MixupMMD, RelaxLoss, and CCL (Liu et al., 2024b). As shown in Figure 5b, our method consistently improves the P1 scores of other defense methods after applying the PAST. The results demonstrate that our method is compatible with existing defense methods and can further improve their performance.

**Ablation on $\ell_1/\ell_2$ regularization and PAST.** To investigate the effectiveness of the adaptive privacy-aware weighting mechanism in PAST, we compare our method with $\ell1/\ell2$ regularization on CIFAR-10 under RMIA. In particular, we fine-tune a trained model using four regularization strategies: $\ell1$ regularization (L1), $\ell1$ regularization + adaptive weight (privacy-aware L1), $\ell2$ regularization (L2), and $\ell2$ regularization + adaptive weight (privacy-aware L2). In Figure 6a, we present the privacy–utility curves to illustrate the trade-offs achieved by different regularization methods. The experimental results show that our PAST (privacy-aware L1) achieves a superior privacy-utility trade-off compared with other defenses. We also present additional results against more attack methods in Appendix G, which show a similar trend. In summary, our experiments highlight the importance of adaptive privacy-aware weighting in PAST for defending against MIAs, showing that assigning unequal penalties to different parameters improves the privacy–utility trade-off.

**How does $\lambda$ affect PAST?** In our PAST, the hyperparameter $\lambda$ in Equation (7) adjusts the importance of the regularization term, similar to the role of $\lambda$ in $\ell1$ regularization. To investigate how $\lambda$ affects the performance of our method, we conduct experiments on CIFAR-10 by fixing $\lambda$ at different values (0.0001, 0.0005, 0.001, 0.002). We illustrate the privacy–utility trade-off under each $\lambda$ by plotting test accuracy against the average attack advantage while varying $\alpha$. As shown in Figure 6b, our method performs better when the value of $\lambda$ is set to 0.0005 and 0.001. When $\lambda$ is too large (e.g., $\lambda = 0.002$), the trade-off cannot achieve high utility. On the other hand, a small value of $\lambda$ (e.g., $\lambda = 0.0001$) results in insufficient regularization, which increases the risk of privacy leakage.

**How does $\alpha$ affect utility and privacy?** We conduct an ablation study to investigate the effect of the hyperparameter $\alpha$ on our method's performance. As shown in Figure 7a, our findings align with insights provided in Section 3.2. As $\alpha$ increases, the regularization increasingly focuses on the small subset of parameters with high privacy sensitivity. On the other hand, a smaller $\alpha$ brings our loss function closer to conventional regularization, increasing privacy risk. Conversely, a larger $\alpha$ leads to stronger regularization of sensitive parameters, resulting in underfitting and lower accuracy. Additionally, we perform experiments to investigate the impact of $\alpha$ on the loss gap in Appendix K. The results show that the loss gap continuously decreases as $\alpha$ increases, indicating that $\alpha$ effectively modulates the weighting of privacy-sensitive parameters.

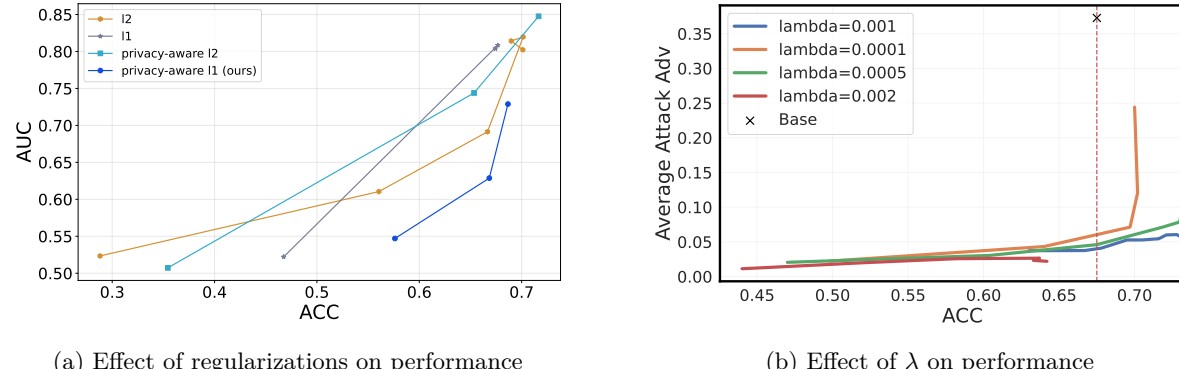

(a) Effect of regularizations on performance

(b) Effect of $\lambda$ on performance

Figure 6: privacy-utility trade-offs on different regularization strategies (a) and different $\lambda$ (b) on CIFAR-10. For different strategies, we first conduct standard training to attain "Base" model, and then tune it with (w/o) applying adaptive weights on $\ell1$ or $\ell2$ regularization. For different $\lambda$, the y-axis represents average attack advantage across various attack methods.

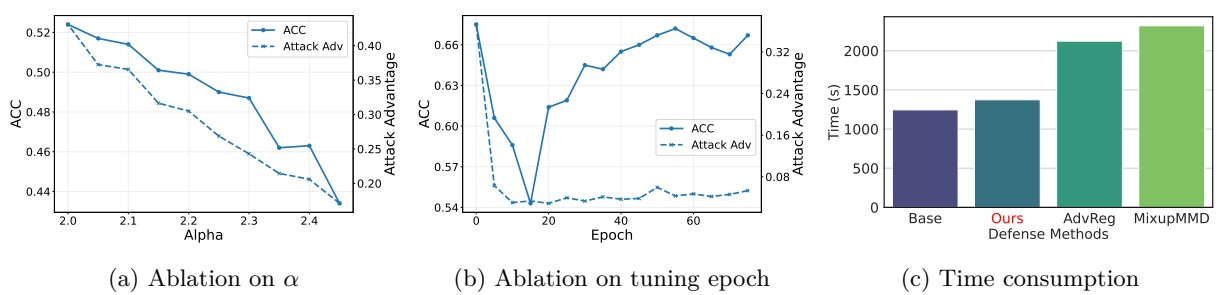

(a) Ablation on $\alpha$      (b) Ablation on tuning epoch      (c) Time consumption

Figure 7: Ablation study of PAST on $\alpha$ and fine-tuning epochs (a & b). Both the attack advantage and accuracy decrease as $\alpha$ increases. For the fine-tuning epoch, the attack performance against PAST stabilizes at a low level after 20 epochs. (c) Computational overhead of various defense methods. The results demonstrate that our method is much more efficient than other defense algorithms.

**How does the number of fine-tuning epochs affect PAST?** In this experiment, we investigate the impact of different sparse training epochs on the privacy-utility trade-off. Specifically, we conduct experiments on CIFAR-10, varying the number of epochs across $[5, 10, \cdots, 75]$. Figure 7b plots the curves of test accuracy and attack advantage over training epochs, where the dotted line denotes attack advantage and the solid line denotes test accuracy. As the number of epochs increases, the attack advantage stabilizes at a low level after 20 epochs, and the classification accuracy starts to stabilize at around 50 epochs. The results show that a few epochs are sufficient to achieve satisfactory defense performance, and more epochs lead to more stable model accuracy. Moreover, we also conduct experiments in Appendix K to explore the impact of different epochs on the loss gap. The results show that only a few epochs significantly decrease the loss gap, demonstrating the effectiveness of our method in reducing the loss gap and improving resistance against membership inference attacks.

**Comparison of computational overhead across defense methods.** The computational overhead of PAST is comparable to standard training, as it incurs no additional expensive computations. It only requires an additional gradient backpropagation step during each tuning epoch to obtain gradients for non-members. In Figure 7c, we report the runtime of various defense methods, where all computation times are measured on DenseNet121 trained on CIFAR-100 using a single RTX 4090 GPU. Compared with standard training, our method incurs only a marginal increase in runtime, while other defense methods introduce substantially greater computational overhead. The results demonstrate that PAST yields significant improvements while incurring only minimal computational overhead.

Table 1: Comparison of test accuracy and attack AUC under PAST defense with unified hyperparameter configurations ($\lambda = 0.001$ and $\alpha = 1.3$) across model architectures.

| Method | ResNet-18 | | MobileViT-S | | DenseNet-121 | |
|---|---|---|---|---|---|---|
| | Test Acc ($\uparrow$) | Attack AUC ($\downarrow$) | Test Acc ($\uparrow$) | Attack AUC ($\downarrow$) | Test Acc ($\uparrow$) | Attack AUC ($\downarrow$) |
| No Defense | 68.370 | 0.719 | 64.020 | 0.713 | 66.320 | 0.754 |
| Ours | 67.380 | 0.535 | 66.940 | 0.527 | 69.750 | 0.543 |

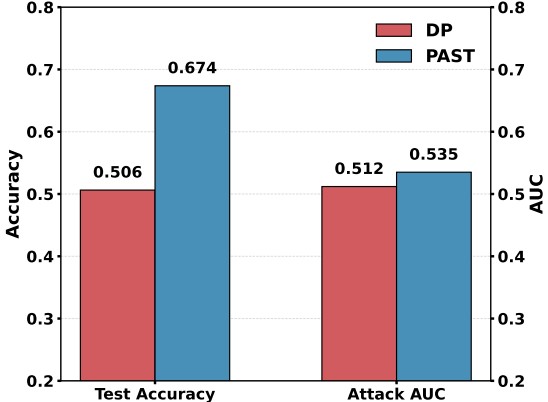

(a) Comparison of test accuracy (utility) and attack AUC (privacy) between PAST and DP-SGD.

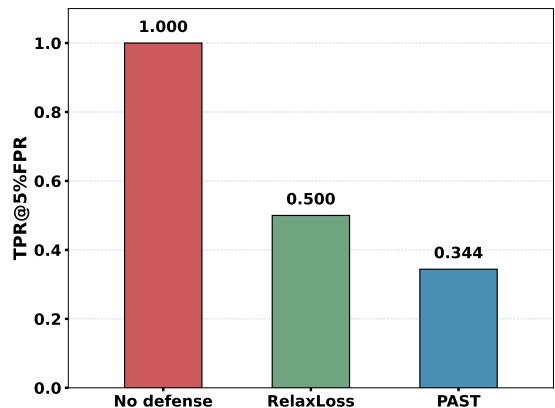

(b) Comparison of TPR@5%FPR of our PAST and RelaxLoss on most vulnerable samples.

**The hyperparameters of PAST are insensitive across different model architectures.** To evaluate the robustness of PAST's hyperparameters across different model architectures, we conduct experiments using a unified hyperparameter configuration ($\lambda = 0.001$ and $\alpha = 1.3$) across three architectures: ResNet-18, MobileViT-S, and DenseNet-121. For each model architecture, we report the test accuracy and attack AUC under no defense and PAST defense settings. As shown in Table 1, under a unified hyperparameter configuration, PAST consistently reduces attack AUC across all three architectures from 0.754 to below 0.550, while preserving test accuracy, demonstrating its effectiveness and robustness across diverse model architectures. Our results indicate that PAST's hyperparameters are insensitive to model architecture, suggesting that our method can be applied across different models with minimal architecture-specific hyperparameter tuning.

## 5  Discussion

**Defense performance comparison between PAST and differential privacy.** Differential privacy is a widely adopted defense method against membership inference attacks, which provides differential privacy guarantees by perturbing per-sample gradients during training via the DP-SGD mechanism (Abadi et al., 2016). To compare the defense performance of our PAST and DP-SGD, we conduct experiments on CIFAR-10 with ResNet-18 in terms of test accuracy and attack AUC under the loss-based attack method. Specifically, we train ResNet-18 using DP-SGD (Abadi et al., 2016) implemented via Opacus (Yousefpour et al., 2021) to obtain a differentially private defended model. To achieve a strong DP-SGD baseline, we conduct an extensive hyperparameter sweep with privacy budgets $\varepsilon \in \{1, 2, 4, 8, 10, 100, 200, 300\}$, gradient clipping thresholds $C \in \{0.5, 1, 2, 5, 10\}$, and $\delta = \in \{10^{-5}, 10^{-6}\}$, resulting in 80 configurations in total. We select the best-performing configuration to ensure a fair and competitive DP-SGD baseline. Notably, while DP-SGD provides provable privacy guarantees, it often incurs a significant utility degradation due to the noise introduced during training.

Figure 8a reports the test accuracy and the corresponding attack AUC of our PAST and DP-SGD, where DP-SGD results are reported at the configuration yielding the highest test accuracy, and PAST results are

Table 2: Attack AUC of gradient-based and loss-based attack methods under PAST defense and no defense on the CIFAR-10 dataset with ResNet-18.

| Method | Loss | Gradient (L1) | Gradient (L2) | Gradient (Mean) | Gradient (Skewness) |
|---|---|---|---|---|---|
| No Defense | 0.719 | 1.000 | 1.000 | 0.981 | 0.695 |
| Ours | 0.535 | 0.492 | 0.491 | 0.796 | 0.682 |

reported under $\lambda = 0.001$, $\alpha = 1.3$. In terms of privacy protection, DP-SGD achieves a lower attack AUC of 0.512, indicating superior resistance to membership inference attacks compared to PAST. In contrast, our PAST achieves a significantly higher test accuracy of 0.674 compared to DP-SGD, but with a slightly higher attack AUC of 0.535. These results indicate that DP-SGD and PAST each exhibit distinct advantages: DP-SGD provides stronger privacy protection but at the cost of substantial utility degradation, whereas PAST achieves superior model utility despite slightly weaker privacy protection.

**Membership privacy evaluation on vulnerable canaries.** Prior work argues that population-level evaluations fail to capture the privacy leakage of the most vulnerable samples, and that membership inference evaluations should also consider worst-case settings rather than average-case settings (Aerni et al., 2024). To investigate worst-case privacy leakage, we construct a set of canaries by selecting the 500 samples most vulnerable to membership inference attacks. Specifically, following previous work (Aerni et al., 2024), we first run membership inference attacks on the undefended model, then select the 500 training samples with the highest membership scores as canaries. We conduct experiments on CIFAR-10 with ResNet-18, comparing our method against baseline defense on an evaluation set comprising the 500 most vulnerable members and 500 non-members.

Our experimental results show that, while our method does not fully eliminate privacy leakage in the worst-case setting, it substantially reduces the risk of exposing the most vulnerable samples. As shown in Figure 8b, PAST substantially reduces worst-case privacy leakage, achieving a TPR@5%FPR of 0.344 compared to 0.500 of RelaxLoss — a 65.6% reduction relative to the undefended model, while preserving a comparable test accuracy (67.38% for PAST and 67.84% for RelaxLoss). Our experiments demonstrate that our method can mitigate membership privacy leakage under the worst-case setting. Additionally, while achieving strong protection in worst-case settings is a promising objective, population-level evaluation remains essential, as preventing privacy leakage across the broader training population is equally important. Overall, the experimental results demonstrate the effectiveness of PAST across diverse evaluation settings.

**Can our method effectively defend against gradient-based attacks?** White-box attacks often pose the most severe threat in membership inference, as they assume full access to all model parameters. Gradient-based attack methods exploit gradient information computed from the model's parameters to distinguish member from non-member data (Salem et al., 2019b). To evaluate whether PAST can defend against white-box attacks, we apply four gradient-based attack methods that compute different statistical metrics on the gradients of the loss with respect to model weights, including L1 norm, L2 norm, mean value, and the skewness of the gradient distribution on the CIFAR-10 dataset with ResNet-18. Additionally, we also include a loss-based attack as a baseline for comparison.

Table 2 reports the attack AUC of gradient-based attacks under no defense and PAST defense, respectively. The experimental results show that our PAST consistently reduces the attack AUC across all attack methods, demonstrating its effectiveness in defending against gradient-based attacks. However, our method exhibits limited defense against attacks based on gradient mean and skewness, where the attack AUC decreases only from 0.981 to 0.796 and from 0.695 to 0.682, indicating that our defense method remains vulnerable to privacy leakage under white-box attack methods. Our findings suggest that, while our defense method can weaken gradient-based attacks, it does not eliminate the risk of privacy leakage. This highlights the necessity for future research focused on developing defenses specifically tailored for white-box attacks.

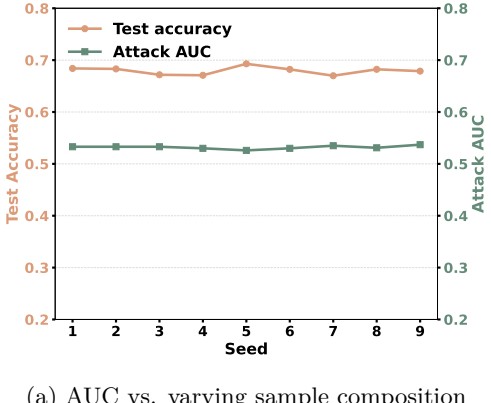 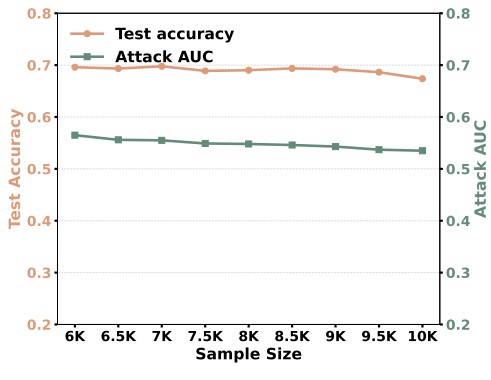

(a) AUC vs. varying sample composition    (b) AUC vs. dataset size

Figure 9: The performance of our method under varying inference set compositions (9a) and data sizes (9b) on CIFAR-10 with ResNet-18.

**How does the inference set affect the performance of our PAST?**    In our work, our PAST applies an adaptive privacy-aware regularization technique to reduce the loss gap between members and non-members. To this end, we introduce an inference set serving as non-member samples to quantify the loss gap between members and non-members. To examine the impact of inference set composition and size on our method's performance, we conduct experiments on the CIFAR-10 dataset with ResNet-18 by separately varying the number of samples and the sample composition of the inference set.

Figure 9a shows the attack AUC and test accuracy of PAST under varying inference set compositions, where different random seeds result in different non-member sample compositions. The results show that both the attack AUC and test accuracy remain consistently stable across different sample compositions, indicating that our method is insensitive to the composition of the inference set. Figure 9b shows the attack AUC and test accuracy of PAST under varying inference set sizes, ranging from 6K to 10K. The results demonstrate a similarly stable trend, confirming that our method can consistently achieve effective defense across different inference set sizes. These findings confirm that PAST is insensitive to both the composition and size of the inference set, consistently achieving reliable defense performance across diverse inference set configurations.

**Is inference set used in PAST vulnerable to membership inference?**    To identify the privacy sensitivity of model parameters, our PAST introduces an inference set as non-member samples to quantify the loss gap between members and non-members. Although our method leverages additional non-member samples during training, it does not incur extra privacy leakage with respect to these samples, as the model never memorizes them. Instead, they serve exclusively as a reference set for computing the loss gap and identifying privacy-sensitive parameters, thereby minimizing the discrepancy between member and non-member losses. To validate this, we conduct membership inference attacks on CIFAR-10 with ResNet-18, where the inference set samples are used as members, to evaluate whether PAST introduces privacy leakage with respect to these samples. Figure 10 shows the attack AUC across three attack scenarios: no defense, using the training set as members, and using the inference set as members. The results show that the no defense baseline achieves an attack AUC of 0.719, whereas attack AUC

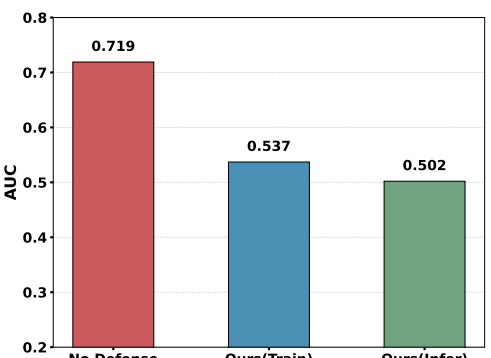

Figure 10: Attack AUC of the loss-based attack method under no defense and PAST defense on CIFAR-10 with ResNet-18.

for the training set and inference set drops to 0.537 and 0.502 under PAST defense. These results demonstrate that the inference set samples are not exposed to membership inference attacks, confirming that PAST does not introduce additional privacy leakage with respect to the inference set.

## 6 Conclusion

In this paper, we first investigate the importance of model parameters in privacy risk and show that only a small fraction of them substantially contribute to privacy leakage. In light of this, we introduce Privacy-aware Sparsity Tuning (PAST), a novel approach that mitigates membership inference attacks by adaptively regularizing model parameters based on their privacy sensitivity. By promoting sparsity among parameters with high privacy sensitivity, the model reduces the loss gap between members and non-members, leading to strong resistance to privacy attacks. Extensive experiments demonstrate that PAST effectively balances privacy and utility, achieving superior performance in the privacy-utility trade-off. Our method is straightforward to implement on trained models and can be incorporated into existing defense methods to further improve their effectiveness. We hope that our insights into privacy-aware regularization inspire further research to explore parameter regularization techniques for enhancing privacy.

**limitations** While PAST demonstrates strong performance on standard classification models such as ResNet and DenseNet, its applicability to large language models remains limited. LLMs are pretrained on massive corpora, encoding rich and complex knowledge across their parameters. The adaptive regularization imposed by PAST may aggressively penalize parameters that are critical to retaining this pretrained knowledge, leading to accuracy degradation. We leave the design of efficient privacy-preserving techniques for LLMs as an important direction for future work. Additionally, our work primarily focuses on gray-box membership inference attacks, which assume that the adversary has access to the model's output logits and predictions. Although our experiments demonstrate the effectiveness of PAST against such attacks, white-box attacks (e.g., gradient-based attacks) may still pose a potential privacy risk, as adversaries can exploit the entire knowledge of the model to bypass existing defenses.

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

# Appendix

## A   Related work

**Over-parameterization in generalization and privacy**   Over-parameterization, where models have significantly more parameters than training examples, has been shown to have a complex relationship with generalization and privacy. While traditional theories suggest that over-parameterization increases overfitting and generalization error, recent research shows that it can sometimes reduce error under certain conditions, such as in high-dimensional ridgeless least-squares problems (Belkin et al., 2020). This phenomenon, known as "double descent", suggests that beyond a critical point, increasing model complexity may lead to better generalization (Belkin et al., 2019; Dar et al., 2021; Hastie et al., 2022). However, from a privacy perspective, over-parameterization has been empirically proven to increase vulnerability to membership inference attacks (Leemann et al., 2023; Dionysiou & Athanasopoulos, 2023). Large language models, in particular, are susceptible to these attacks, with attackers able to extract sensitive training data (Carlini et al., 2021; Mireshghallah et al., 2022). Theoretical evidence also indicates that there is a clear parameter-privacy trade-off, where an increase in the number of parameters amplifies the privacy risks by enhancing model memorization (Yeom et al., 2018; Tan et al., 2022). Consequently, while over-parameterization can sometimes improve generalization, its impact on privacy remains a significant concern, especially in the context of membership inference attacks.

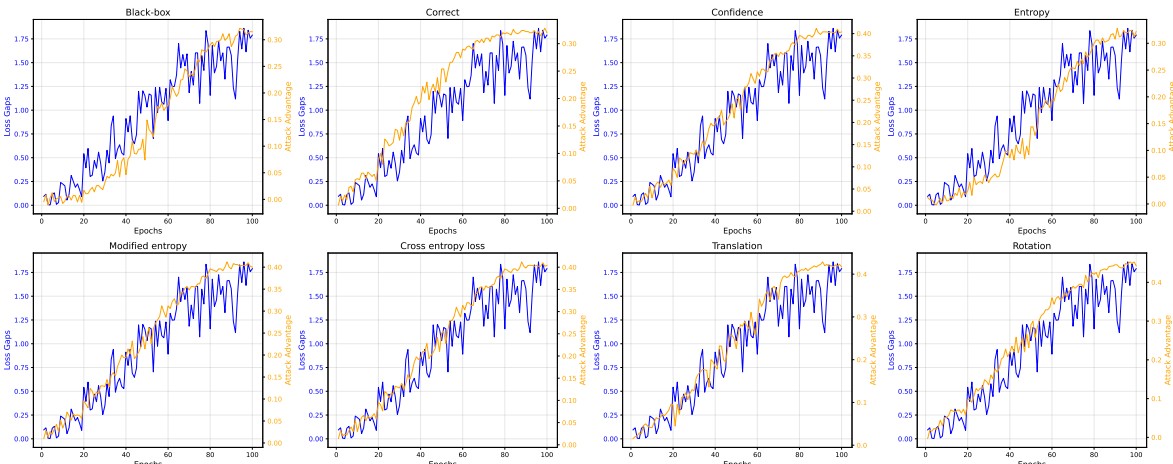

Figure 11: The variation of the loss gap and the attack advantage across various attacks during standard training. During standard training, the loss gap synchronously increases with the attack advantage across various attacks. This indicates that the loss gap, as a privacy proxy, can effectively capture the effects of various attacks.

**Over-parameterization in MIA defenses** To mitigate the privacy risks associated with over-parameterization, several defense mechanisms have been proposed. One effective approach is network pruning, where unnecessary parameters are removed to reduce model complexity. Research shows that pruning not only preserves utility but also significantly reduces the risk of privacy leakage, in scenarios including MIA (Huang et al., 2020; Wang et al., 2021) and Unlearning (Hooker et al., 2019; Wang et al., 2022; Ye et al., 2022; Liu et al., 2024a). Additionally, techniques combining pruning with federated unlearning have demonstrated effectiveness in protecting privacy by selectively forgetting specific data during the training process (Wang et al., 2022). Regularization methods, such as $\ell 2$ regularization (Kaya et al., 2020), sparsification (Bagmar et al., 2021) and dropout (Galinkin, 2021), also play a critical role in defending against MIAs by discouraging the model from overfitting to training data. Interestingly, while over-parameterization generally increases privacy risks, when paired with appropriate regularization, it can maintain both utility and privacy (Tan et al., 2023). Furthermore, studies indicate that initialization strategies and ensemble methods can further alleviate privacy risks on over-parameterized model (Rezaei et al., 2021; Ye et al., 2024). These techniques illustrate that even in overparameterized models, privacy risks can be mitigated through careful design, preserving the balance between utility and privacy.

## B    Rationale and generalizability of the loss gap as a privacy proxy.

To clarify the rationale for using the loss gap as a proxy risk for privacy, we theoretically characterize that the loss gap is positively correlated with the attack advantage in metric-based attacks that uses the loss.

**Proposition B.1.** *Let $\epsilon$ be a random variable denoting loss, such that $\epsilon \sim N(\mu_S, \sigma_S^2)$ when $m = 1$ and $\epsilon \sim N(\mu_D, \sigma_D^2)$ when $m = 0$. Then the loss gap $(\mu_D - \mu_S)$ is positively correlated with the attack advantage, defined in Equation (8).*

*Proof.* The membership advantage of $\mathcal{A}_{\text{loss}}$ is (as defined in Equation (8)):

$$Adv = \Pr(\mathcal{A} = 1 | m = 1) - \Pr(\mathcal{A} = 1 | m = 0) \tag{10}$$

$$= \Pr(\epsilon \leqslant \tau | m = 1) - \Pr(\epsilon \leqslant \tau | m = 0) \tag{11}$$

$$= \Phi\left(\frac{\tau - \mu_S}{\sigma_S}\right) - \Phi\left(\frac{\tau - \mu_D}{\sigma_D}\right) \tag{12}$$

where $\Phi(\cdot)$ is the cumulative distribution function of standard normal distribution. Note that $\Pr(\mathcal{A} = 1 | m = 0)$ is false positive rates of the adversary, which is expected to be controlled at a small value (Leemann et al.,

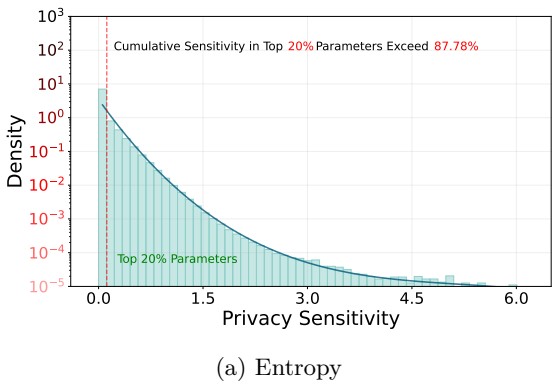
(a) Entropy

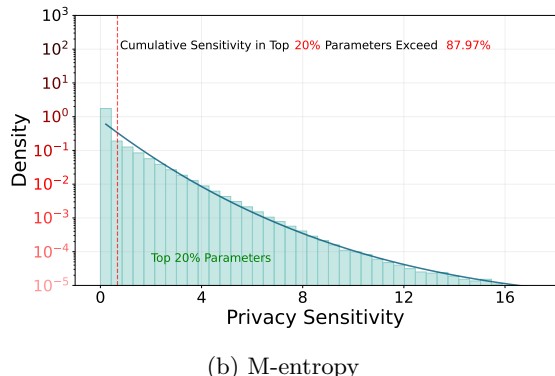
(b) M-entropy

Figure 12: The distribution of privacy sensitivity across all parameters computed from the entropy gap (a) and the M-entropy gap (b). Similar to privacy sensitivity computed from the loss gap, only a few parameters have large privacy sensitivity.

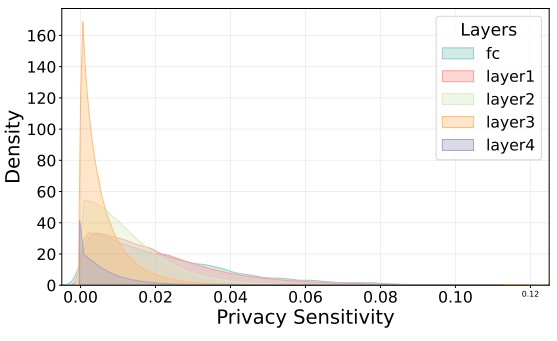
(a) Sensitivity distribution of different layers

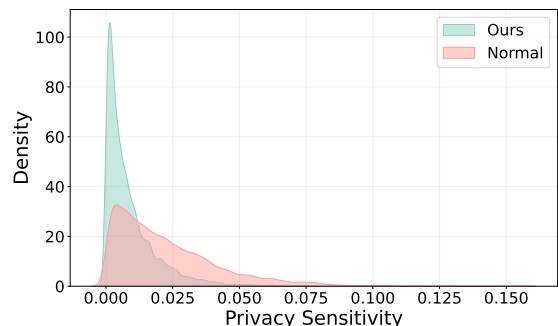
(b) Sensitivity distribution of Normal and PAST

Figure 13: (a) The sensitivity distribution of parameters across all layers. Within each layer, the finding that "only a small fraction of parameters substantially impacts privacy risk" remains significant. (b) The sensitivity distribution of fc layer parameters for standard training (Normal) and PAST. PAST has fewer privacy-sensitive parameters, indicating that it remains effective even if privacy migration occurs, as the overall sensitivity is reduced.

2023; Tan et al., 2022). Assume $\tau$ is chosen such that $\Phi(\frac{\tau - \mu_D}{\sigma_D}) = \alpha$, then we have:

$$Adv = \Phi\{\frac{\Phi^{-1}(\alpha)\sigma_D + \mu_D - \mu_S}{\sigma_S}\} - \alpha \tag{13}$$

Since $\frac{\partial (Adv)}{\partial (\mu_D - \mu_S)} = \frac{1}{\sigma_S}\phi\{\frac{\Phi^{-1}(\alpha)\sigma_D + \mu_D - \mu_S}{\sigma_S}\} > 0$, this implies that the loss gap $(\mu_D - \mu_S)$ is positively correlated with the attack advantage $Adv$. □

For more general attack methods, we refer to previous work (Sablayrolles et al., 2019b) and demonstrate that the optimal ones rely solely on the loss.

From the Bayesian viewpoint, we know the posterior distribution follows (detailed proof in Appendix C):

$$\mathbb{P}(\theta \mid z_1, \ldots, z_n) \propto e^{-\sum_{i=1}^n \ell(\theta, z_i)}. \tag{14}$$

Assume that binary membership variables $m_1, m_2, \ldots, m_n$ are drawn independently, with probability $\lambda = P(m_i = 1)$. Then Equation (14) can be rewritten as

$$\mathbb{P}(\theta \mid z_1, \ldots, z_n, m_1, \ldots, m_n) \propto e^{-\sum_{i=1}^n m_i \ell(\theta, z_i)}. \tag{15}$$

Taking the case of $z_1$ without loss of generality, membership inference determines, given parameters $\theta$ and sample $z_1$, whether $m_1 = 1$ or $m_1 = 0$.

**Definition B.2.** (Membership inference). *Inferring the membership of sample $z_1$ to the training set amounts to computing:*

$$\mathcal{M}(\theta, z_1) := \mathbb{P}(m_1 = 1 | \theta, z_1). \tag{16}$$

**Notation.** We denote by $\sigma$ the sigmoid function $\sigma(u) = (1 + e^{-u})^{-1}$. We collect the knowledge about the other samples and their memberships into the set $T = \{z_2, \ldots, z_n, m_2, \ldots, m_n\}$. And we denote by $p_{\mathcal{T}}(\theta)$ the posterior over the parameters given samples $z_2, \ldots, z_n$ and memberships $m_2, \ldots, m_n$:

$$p_{\mathcal{T}}(\theta) := \frac{e^{-\sum_{i=2}^{n} m_i \ell(\theta, z_i)}}{\int_t e^{-\sum_{i=2}^{n} m_i \ell(t, z_i)} dt}. \tag{17}$$

**Theorem B.3.** *Given a parameter $\theta$ and a sample $z_1$, the optimal membership inference is given by:*

$$\mathcal{M}(\theta, z_1) = \mathbb{E}_{\mathcal{T}} \left[ \sigma \left( s(z_1, \theta, p_{\mathcal{T}}) + t_\lambda \right) \right]. \tag{18}$$

*where we define:*

$$\tau_p(z_1) := -\log \left( \int_t e^{-\ell(t, z_1)} p(t) dt \right) \tag{19}$$

$$s(z_1, \theta, p) := (\tau_p(z_1) - \ell(\theta, z_1)) \tag{20}$$

$$t_\lambda := \log(\frac{\lambda}{1 - \lambda}) \tag{21}$$

*Proof.* By the law of total expectation, we have:

$$\mathcal{M}(\theta, z_1) = \mathbb{P}(m_1 = 1 \mid \theta, z_1) \tag{22}$$

$$= \mathbb{E}_{\mathcal{T}}[\mathbb{P}(m_1 = 1 \mid \theta, z_1, \mathcal{T})]. \tag{23}$$

Applying Bayes' formula:

$$\mathbb{P}(m_1 = 1 \mid \theta, z_1, \mathcal{T}) = \frac{\mathbb{P}(\theta \mid m_1 = 1, z_1, \mathcal{T})\mathbb{P}(m_1 = 1)}{\mathbb{P}(\theta \mid z_1, \mathcal{T})} \tag{24}$$

$$= \frac{\alpha}{\alpha + \beta} = \sigma \left( \log \left( \frac{\alpha}{\beta} \right) \right), \tag{25}$$

where:

$$\alpha := \mathbb{P}\left( \theta \mid m_1 = 1, z_1, \mathcal{T} \right) \mathbb{P}(m_1 = 1) \tag{26}$$

$$\beta := \mathbb{P}\left( \theta \mid m_1 = 0, z_1, \mathcal{T} \right) \mathbb{P}(m_1 = 0) \tag{27}$$

Singling out $m_1$ in Equation (15) yields the following expressions for $\alpha$ and $\beta$:

$$\alpha = \lambda \frac{e^{-\ell(\theta, z_1)} e^{-\sum_{i=2}^{n} m_i \ell(\theta, z_i)}}{\int_t e^{-\ell(t, z_1)} e^{-\sum_{i=2}^{n} m_i \ell(t, z_i)} dt} \tag{28}$$

$$= \lambda \frac{e^{-\ell(\theta, z_1)} p_{\mathcal{T}}(\theta)}{\int_t e^{-\ell(t, z_1)} p_{\mathcal{T}}(t) dt}, \tag{29}$$

and

$$\beta = (1 - \lambda) \frac{e^{-\sum_{i=2}^{n} m_i \ell(\theta, z_i)}}{\int_t e^{-\sum_{i=2}^{n} m_i \ell(t, z_i)} dt} = (1 - \lambda) p_{\mathcal{T}}(\theta). \tag{30}$$

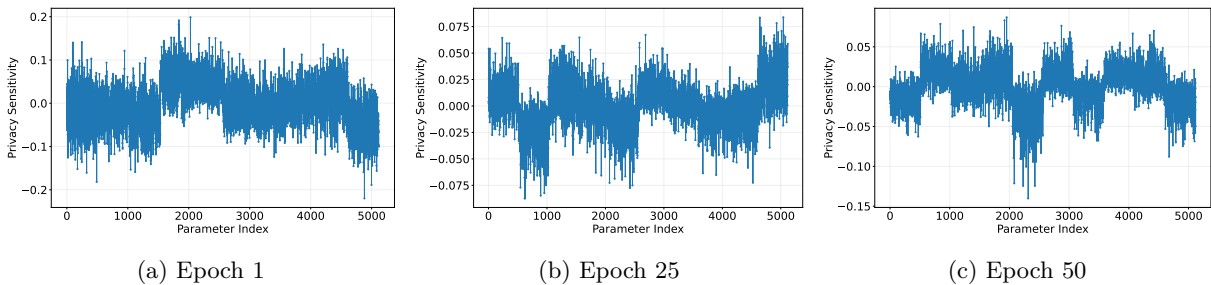

(a) Epoch 1          (b) Epoch 25          (c) Epoch 50

Figure 14: The change of privacy sensitivity distribution during PAST. We fixed the x-axis as parameter index and the y-axis as privacy sensitivity, reporting results for epochs 1, 25, and 50. It can be observed that privacy sensitivity indeed migrates, while the overall privacy sensitivity decreases over time.

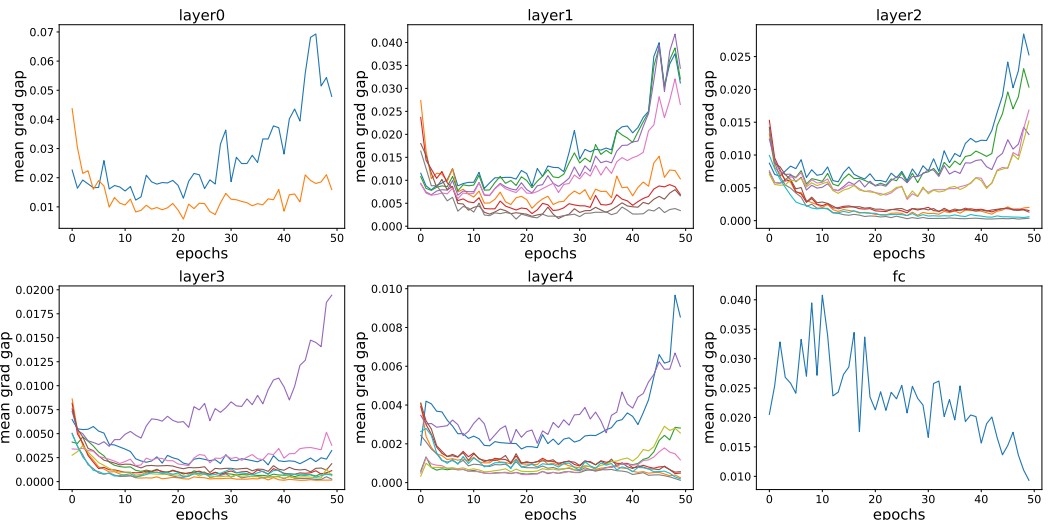

Figure 15: Variation of privacy sensitivity for each module grouped by layers during PAST. Each line indicates the mean privacy sensitivity of a module, and each subfigure indicates a specific layer.

Thus,

$$\log\left(\frac{\alpha}{\beta}\right) = -\ell(\theta, z_1) - \log\left(\int_t e^{-\ell(t, z_1)} p_{\mathcal{T}}(t) dt\right) + t_\lambda \tag{31}$$

$$= s(z_1, \theta, p_{\mathcal{T}}) + t_\lambda, \tag{32}$$

which gives the expression for $\mathcal{M}(\theta, z_1)$.      □

It can be observed in Theorem B.3 that $\mathcal{M}(\theta, z_1)$ does not depend on the parameters $\theta$ beyond the evaluation of the loss $\ell(\theta, \cdot)$. This means that if we can compute $\mathcal{T}_p$ or approximate it well enough, then the optimal membership inference depends only on the loss. In terms of the generalizability of using the loss gap as a proxy for privacy risk, we also empirically demonstrate the effectiveness of the loss gap as a privacy proxy by showing its relationship with the attack advantage across various attack methods. Specifically, in Figures 1a and 11, we observe that during standard training, the loss gap synchronously increases with the attack advantage, suggesting that the loss gap can capture different aspects of the model's output and behavior.

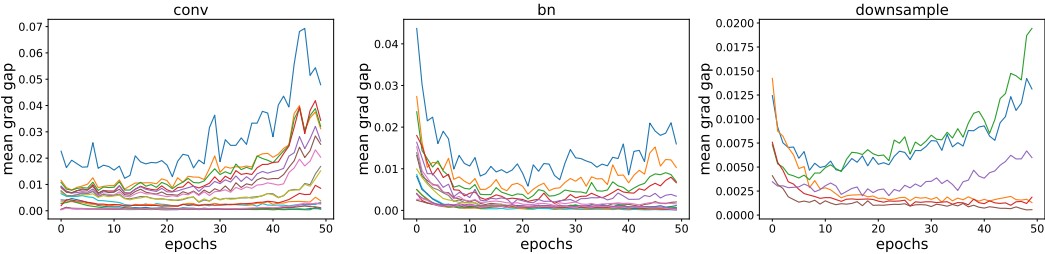

Figure 16: Variation of privacy sensitivity for each module grouped by module types during PAST. Each line indicates the mean privacy sensitivity of a module, and each subfigure indicates a specific module type.

## C   Bayesian posterior distribution

The canonical form of (supervised) deep learning is that of empirical risk minimization. Following the notation in Section 2, it can be expressed as:

$$\theta_{\mathrm{MAP}} = \arg\min_{\theta \in \mathbb{R}^S} \mathcal{L}(\mathcal{S}; \theta) = \arg\min_{\theta \in \mathbb{R}^S} \left( R(\theta) + \sum_{n=1}^{N} \ell(x_n, y_n; \theta) \right). \tag{33}$$

From the Bayesian viewpoint, these terms can be identified with i.i.d. log-***likelihoods*** and a log-***prior***, respectively and, thus, $\theta_{MAP}$ is indeed a ***maximum a-posteriori (MAP)*** estimate:

$$\ell(x_n, y_n; \theta) = -\log p(y_n \mid f_\theta(x_n)) \quad \text{and} \quad R(\theta) = -\log p(\theta). \tag{34}$$

For example, the widely used weight regularizer $R(\theta) = \frac{1}{2}\gamma^{-2}||\theta||^2$ (a.k.a. weight decay) corresponds to a centered Gaussian prior $p(\theta) = \mathcal{N}(\theta; 0, \gamma^2 I)$, and the cross-entropy loss amounts to a categorical likelihood. Hence, the exponential of the negative training loss $exp(-\mathcal{L}(\mathcal{D}; \theta))$ amounts to an ***unnormalized posterior***. By normalizing it, we obtain

$$p(\theta \mid \mathcal{S}) = \frac{1}{Z} p(\mathcal{S} \mid \theta) p(\theta) = \frac{1}{Z} \exp(-\mathcal{L}(\mathcal{S}; \theta)), \quad Z := \int p(\mathcal{S} \mid \theta) p(\theta) d\theta. \tag{35}$$

## D   Additional results for privacy sensitivity distribution

### D.1   Privacy sensitivity distribution based on other metrics.

To verify whether the findings in motivation (Figure 1b, *Most parameters contribute only marginally to the privacy risk*) are generalizable, we plot the privacy sensitivity based on alternative metrics. Specifically, we choose the Entropy gap and M-entropy gap, as methods similar to the loss gap can be applied to these metrics to obtain the privacy sensitivity of each parameter. Concretely, we compute the Entropy (Salem et al., 2019a) and M-entropy (Song & Mittal, 2021) for both member and non-member data, then calculate the Entropy gap and M-entropy gap. Finally, we take the derivatives with respect to each parameter to obtain the privacy sensitivity of each model parameter. The distributions of privacy sensitivity based on the Entropy gap and M-entropy gap are shown in Figure 12a and Figure 12b, respectively. It can be observed that the finding, "Most parameters contribute marginally to the privacy risk," still holds.

### D.2   Privacy sensitivity within each layer

In this part, we provide empirical evidence that the phenomenon—where only a small fraction of parameters substantially affects privacy risk—exists within each layer of the neural network, rather than being solely attributable to differences between layers. Using a ResNet18 model trained on CIFAR-10, we present the privacy sensitivity distribution of parameters within each layer, as shown in Figure 13a. The observation that

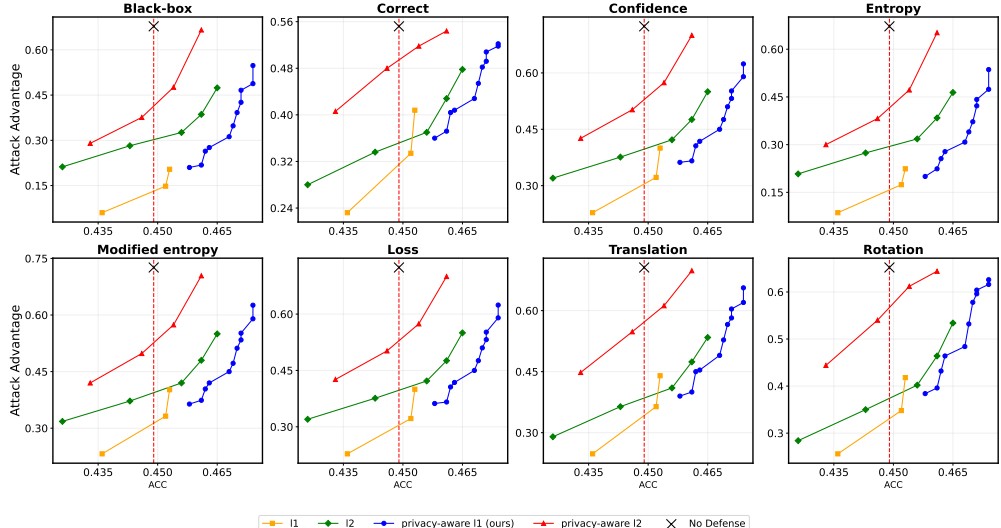

Figure 17: privacy-utility trade-offs of different epochs on CIFAR-10. Each subplot is allocated to a distinct attack method, wherein individual curves represent the performance of a defense mechanism under different hyperparameter settings. The horizontal axis represents the target models' test accuracy, where a large value is preferred, and the vertical axis represents the corresponding attack advantage (defined in Definition 8), where a small value is preferred. To underscore the disparity between the defense methods and the vanilla (undefended model), we plot the dotted line originating from the vanilla results.

only a small fraction of parameters have large privacy sensitivities holds within each layer. This indicates that the finding is not merely due to parameters closer to the output layer having more influence on gradients and results.

### D.3    Does privacy migrate among parameters?

We analyze the changes in privacy sensitivity during the PAST training process and confirmed the existence of privacy migration. Specifically, we plot the distribution of privacy sensitivity against parameter index at the 1st, 25th, and 50th epochs during training on the fully connected layer of ResNet18 trained on CIFAR-10. The results, shown in Figure 14, confirm that privacy migrates from heavily regularized parameters to others during PAST.

The migration of privacy sensitivity does not affect the effectiveness of our method. Our proposed regularization focuses on parameters deemed privacy-sensitive at each epoch, and these parameters can change dynamically. As shown in Figure 13b, we compare the privacy sensitivity distributions of the FC layer for models trained with PAST versus standard training. The results indicate a significant reduction in the overall privacy sensitivity (with the mean reduced from 0.0223 to 0.0088), demonstrating the robustness of our method in mitigating privacy risks.

### D.4    For which layers and modules are more effective?

We illustrate the average privacy sensitivity dynamics during the PAST process across different modules (eg., fc layer). Taking the ResNet18 model as an example, we plot the variation in Figure 15 and Figure 16. Each line indicates the mean privacy sensitivity of a module. In Figure 15 and Figure 16, results are grouped by layers and module types, respectively. It can be observed that the deeper layers are more effective than the earlier ones, and batch normalization (BN) and linear layers contribute more significantly than convolutional layers. Notably, the loss gap of all convolutional layers in the third and fourth blocks almost stabilizes at zero.

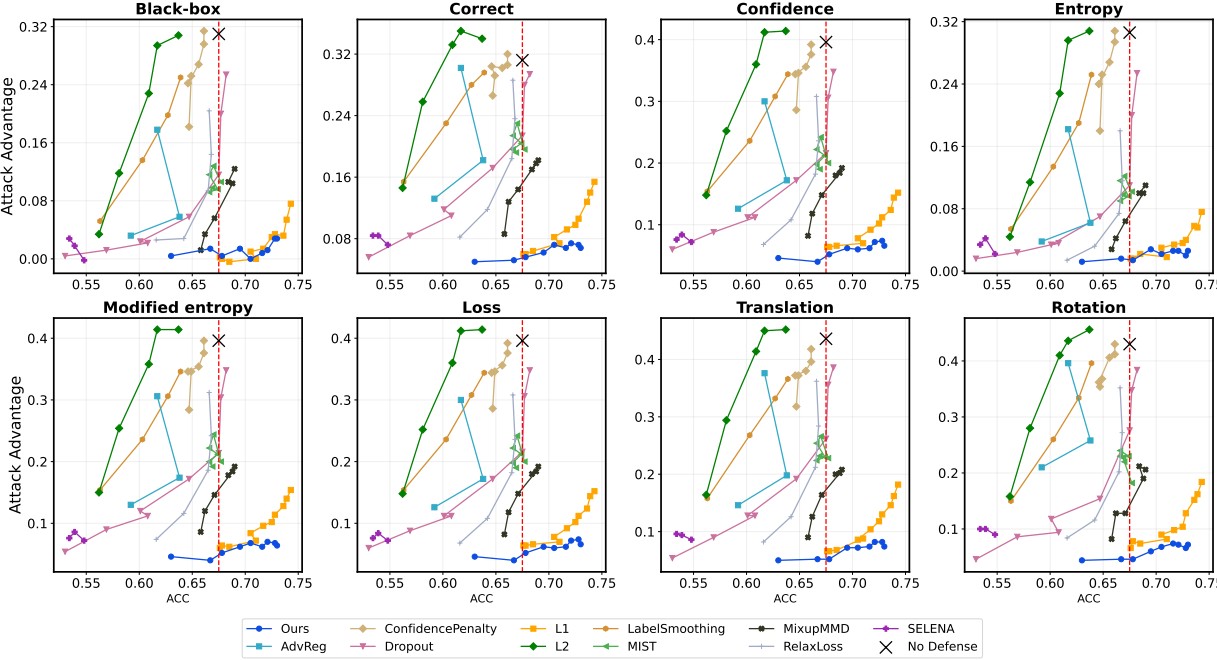

Figure 18: Comparisons of various defense mechanisms on CIFAR-10 dataset utilizing ResNet18 architecture. Each subplot is allocated to a distinct attack method, wherein individual curves represent the performance of a defense mechanism under different hyperparameter settings. The horizontal axis represents the target models' test accuracy (larger values are better), and the vertical axis represents the corresponding attack advantage (defined in Definition 8, smaller values are better). To underscore the disparity between the defense methods and the vanilla (undefended model), we plot the dotted line originating from the vanilla results.

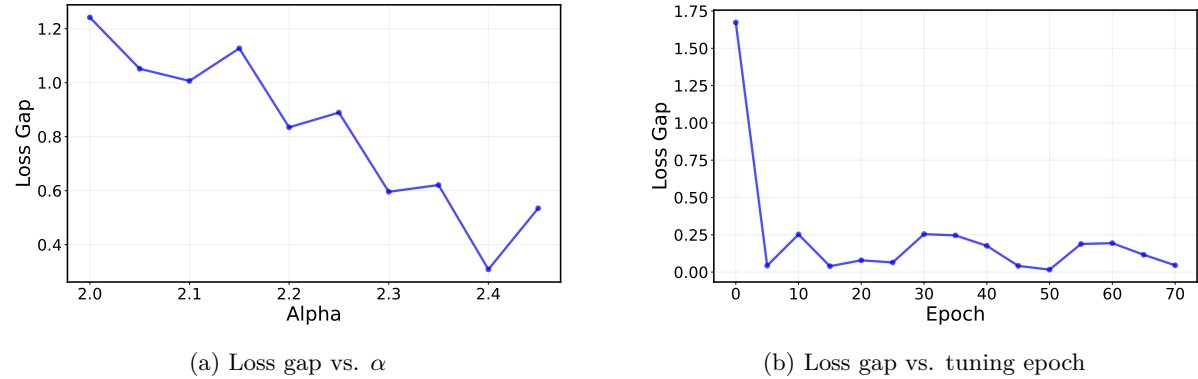

(a) Loss gap vs. $\alpha$          (b) Loss gap vs. tuning epoch

Figure 19: (a) Comparisons of the loss gap between members and non-members under different $\alpha$ values. The loss gap continuously decreases as $\alpha$ increases. (b) The variation of the loss gap under different epochs. As the number of epochs increases, the loss gap decreases rapidly, reaching nearly stability after just 5 epochs. This demonstrates that PAST's effect in shrinking the loss gap can be achieved with only a few epochs.

# E   Definition of Gini index

The Gini index is a widely used metric for quantifying the sparsity of a distribution (Hurley & Rickard, 2009; Goswami et al., 2021), where a larger Gini index indicates a higher degree of sparsity. Given a vector

$\vec{c} = [c_1, c_2, \ldots, c_N]$, the Gini index $S(\vec{c})$ is defined as:

$$S(\vec{c}) = 1 - 2\sum_{k=1}^{N} \frac{c_{(k)}}{\|\vec{c}\|_1} \left(\frac{N - k + \frac{1}{2}}{N}\right)$$

where $c_{(k)}$ denotes the $k$-th smallest element of $\vec{c}$ and $|\vec{c}|_1$ represents the $\ell_1$ norm of $\vec{c}$.

In our experiments (e.g., Figure 2b), we observe that our method consistently increases the Gini index of the model parameters, highlighting its effectiveness in influencing important parameters (i.e., those with high privacy sensitivity).

## F    Hyperparameters settings

This section details the hyperparameter settings used in our experiments, including the regularization strength $\lambda$ and the parameter-specific scaling factor $\alpha$. We also describe the hyperparameter tuning process.

### F.1    Hyperparameter values

Table 3 presents the values of $\lambda$ and the range of $\alpha$ for each dataset and model configuration. The hyperparameter $\lambda$ controls the overall regularization strength, while $\alpha$ modulates the regularization intensity across different model parameters based on their privacy sensitivity.

Table 3: Hyperparameter settings for $\lambda$ and $\alpha$ across datasets and models.

| Dataset | Model | $\lambda$ | Range of $\alpha$ |
|---------|-------|-----------|-------------------|
| CIFAR-100 | DenseNet-121 | 0.0001 | [2.0, 2.4] |
| CIFAR-10 | DenseNet-121 | 0.001 | [1.3, 1.7] |
| | ResNet-18 | 0.001 | [1.1, 1.7] |
| | SqueezeNet | 0.0003 | [1.7, 1.9] |
| | MobileViT-S | 0.001 | [1.6, 2.0] |

### F.2    Hyperparameter tuning

To determine suitable hyperparameter values, we first tune $\lambda$ to optimize model accuracy while fixing $\alpha$ at a small constant (e.g., 1.0). Subsequently, we evaluate the privacy-utility trade-off by varying $\alpha$ within the specified ranges. Unless otherwise specified, we apply this procedure to each model and dataset independently. A sensitivity analysis of $\alpha$, presented in Figure 6b, demonstrates that our method's performance is robust to variations in $\lambda$, indicating low sensitivity to this hyperparameter.

## G    Additional ablation on $\ell_1/\ell_2$ regularization and PAST

In Section 4.2, only the RMIA performance on CIFAR-10 is shown due to layout constraints. Here in Figure 17, we supplement the detailed results of the ablation study under different attack methods on CIFAR-100. It can be observed that the performance varies under different attack methods, but the overall privacy-utility trade-off of PAST evidently surpasses others.

## H    Comparison with various defenses on CIFAR-10

In this section, we compare the defense performance of our proposed PAST with various existing defense methods on CIFAR-10. We report the privacy-utility trade-offs by plotting the curve of attack advantage versus test accuracy in Figure 18, where lower attack advantage and higher test accuracy are desirable. The detailed experimental settings are described in Section 4.1. Overall, PAST still consistently outperforms other defenses.

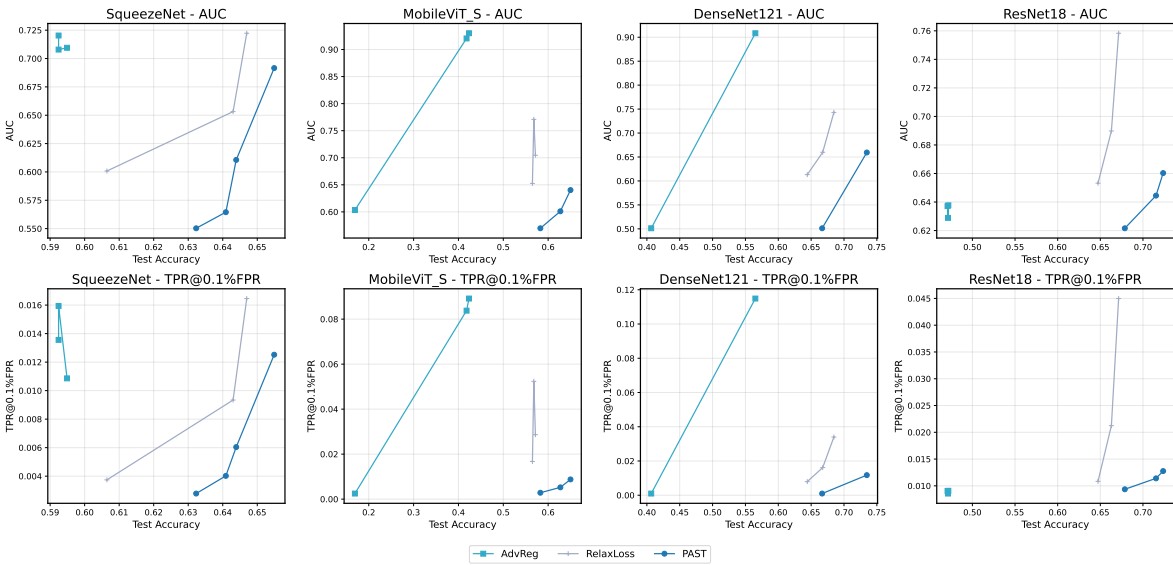

Figure 20: AUC and TPR@0.1%FPR on CIFAR-10 with different models, including SqueezeNet, Mobile-ViT_S, DenseNet121 and ResNet18. We present results from the RMIA attack, showing that our PAST consistently outperforms other defense methods.

Table 4: Comparison of test accuracy, attack AUC, and P1 score under PAST defense with unified hyper-parameter configurations ($\lambda = 0.001$ and $\alpha = 1.3$) across different seeds.

| Metric | Seed 1 | Seed 2 | Seed 3 | Seed 4 | Seed 5 | Seed 6 | Seed 7 | Seed 8 | Seed 9 | Mean $\pm$ Std |
|---|---|---|---|---|---|---|---|---|---|---|
| Test Acc (%) | 68.40 | 68.31 | 67.18 | 67.08 | 69.29 | 68.22 | 66.99 | 68.23 | 67.88 | **67.95 $\pm$ 0.74** |
| Attack AUC | 0.533 | 0.533 | 0.533 | 0.530 | 0.526 | 0.530 | 0.535 | 0.531 | 0.537 | **0.532 $\pm$ 0.003** |
| P1 Score | 0.795 | 0.794 | 0.786 | 0.786 | 0.802 | 0.793 | 0.783 | 0.794 | 0.790 | **0.791 $\pm$ 0.006** |

# I  Results across various model architectures

In this section, we demonstrate the strong generalization performance of PAST across various models. Specifically, we conduct experiments on DenseNet121, ResNet18, SqueezeNet, and MobileViT_S with CIFAR-10, where MobileViT_S adopts a transformer architecture while the other three are CNN-based architectures. We present the results against RMIA on CIFAR-10 under different model architectures in Figure 20. The experimental settings are consistent with inference-based attacks described in Section 4.1. As shown in Figure 20, the trade-offs between utility and the two privacy metrics — attack AUC and TPR@0.1%FPR — are consistently better than those of other defense methods, indicating that the superiority of PAST can generalize to various model architectures.

# J  Stability Analysis across Random Seeds

To evaluate the performance of our PAST across different random seeds, we report statistical metrics on CIFAR-10 with ResNet-18 across varying seeds. The experimental results in Table 4 show that our PAST achieves a mean P1 score of 0.791 with a standard deviation of 0.006 across 9 random seeds. The small standard deviations indicate that the improvements achieved by PAST are stable across different random seeds, rather than a result of random chance.

## K   Loss gap for different epochs and $\alpha$ value

In Figure 7a and Figure 7b, the ablation on tuning the epoch and $\alpha$ only reports the changes in utility (test accuracy) and privacy (attack advantage). Here, we supplement with changes in the privacy proxy loss gap as $\alpha$ and epoch vary. Figures 19a and 19b are consistent with our finding that the loss gap effectively reflects privacy risk and can be used during training to identify privacy-sensitive parameters.

