# OpenReview forum: "Defending Membership Inference Attacks via Privacy-aware Sparsity Tuning"
_TMLR — Under review for TMLR_

### Review · Reviewer_QcfL · 2026-04-11

**Summary Of Contributions:**

This paper studies defenses against membership inference attacks (MIAs) through parameter regularization. The core observation is that model parameters do not contribute equally to privacy leakage: the paper argues empirically that only a small subset of parameters are highly privacy-sensitive, while most have relatively small impact on leakage. Based on this observation, the authors propose Privacy-Aware Sparsity Tuning (PAST), a fine-tuning method that applies parameter-specific ℓ1-style regularization weights based on the gradient of a member/non-member loss gap proxy. The goal is to selectively sparsify privacy-sensitive parameters while preserving utility-critical ones.

The paper provides:
(1) an empirical analysis showing concentration of privacy sensitivity in a small subset of parameters;
(2) a concrete adaptive regularization method with weighting controlled by privacy sensitivity and a focusing exponent 𝛼; and
(3) experiments across tabular and image datasets, multiple architectures, and a broad set of attacks, including LiRA and RMIA, showing improved privacy-utility trade-offs over several baselines. The method is also presented as a plug-in fine-tuning stage that can improve pre-existing defenses.

Key strengths: the paper is well motivated, the method is simple and practically deployable, the experimental evaluation is broad, and the fine-tuning formulation makes the defense easy to combine with existing methods.
Key weaknesses: the main conceptual dependence on the loss-gap proxy remains only partially justified; the evaluation is broad but still limited to classification and the assumed MIA setting; and several important robustness/generalization questions remain open, such as sensitivity to the member/non-member reference sets, transfer to other privacy notions, and comparison to stronger privacy-preserving training paradigms such as differential privacy.

**Additional Comments:**

N/A

**Audience:**

Yes

**Audience Explanation:**

Membership inference remains a central privacy problem in machine learning, and this paper contributes a practically appealing defense that is simple, modular, and compatible with standard training pipelines.

**Claims And Evidence:**

Yes

**Claims Explanation:**

The empirical evidence is generally strong and clearly presented. The paper evaluates PAST on five datasets (Texas100, Purchase100, CIFAR-10, CIFAR-100, and ImageNet), multiple architectures, and a wide collection of attacks spanning neural, metric-based, augmentation-based, and inference-based MIAs. It also compares against a substantial list of baselines, including MIST, SELENA, RelaxLoss, Mixup+MMD, adversarial regularization, dropout, label smoothing, confidence penalty, and standard ℓ1/ℓ2 regularization. Across these evaluations, the privacy-utility curves and summary metrics consistently favor PAST.

I also found the ablations reasonably convincing. The paper studies the effects of 𝜆,α, fine-tuning epochs, and the adaptive weighting mechanism itself. In particular, the comparison between privacy-aware and non-privacy-aware ℓ1/ℓ2 regularization helps isolate the contribution of the adaptive weighting rather than attributing gains merely to extra fine-tuning or sparsification. The efficiency discussion is also useful, since the method appears to incur only modest extra cost relative to standard training.

That said, I do not think every claim is equally strong. The central design choice is to use the member/non-member loss gap as a proxy for privacy leakage and then differentiate it with respect to parameters. The paper gives both empirical motivation and some theoretical support for this proxy, which is helpful, but the overall argument is still somewhat indirect because the defense is optimized against a surrogate rather than against attack success directly. The reported results suggest this surrogate works well in the studied settings, but the paper would be stronger with a clearer account of when this proxy may fail or become misaligned with stronger or adaptive attacks.

A second limitation is external validity. The experiments are extensive within the image/tabular classification setting, but they do not yet establish that the same mechanism would remain effective for other problem classes, larger modern models outside the tested regimes, or other attacker capabilities. So, while I find the evidence convincing for the paper’s empirical scope, I would slightly temper the generality of the strongest claims.

**Requested Changes:**

1. Strengthen the discussion around the loss-gap proxy. The paper should more clearly articulate the limits of optimizing privacy sensitivity via the member/non-member loss gap rather than directly against attacks. In particular, it would help to discuss cases where this proxy may not correlate well with attack success, and whether adaptive attackers could exploit signals not captured by this objective.

2. Clarify the exact assumptions needed to compute PAST in practice. The method uses member and non-member subsets 𝑆𝑚 and 𝑆𝑛, and the experiments rely on an inference set. The paper should explain more explicitly how these sets are chosen, how sensitive results are to their size/composition, and what practitioners should do when an appropriate non-member reference set is not naturally available.

3. Better delimit the scope of the claims. The current presentation can read as broadly applicable, but the evidence is concentrated on supervised classification with specific datasets and architectures. The paper should more explicitly separate demonstrated empirical scope from broader conjectures.

4. Add more analysis of robustness across hyperparameters and model states. The paper already studies 𝛼, 𝜆, and epochs, but it would be helpful to report how stable the method is across random seeds and whether the optimal fine-tuning budget varies substantially across datasets/models.

5. Include a more direct comparison or discussion relative to stronger privacy-preserving training paradigms, especially differential privacy. Even if the goal is not to compete with DP, readers would benefit from a clearer positioning of what guarantees PAST does and does not provide.

6. Improve presentation clarity around the method definition. The normalization of privacy sensitivities within modules and the role of 𝛼 are important to the method, but the intuition could be explained more explicitly, especially why module-wise normalization is preferable and how it affects different architectures.

---

### Review · Reviewer_EDpQ · 2026-04-29

**Summary Of Contributions:**

This paper introduces a defense technique against membership inference attacks based on weighted $\ell_1$ regularization of model parameters. Rather than applying uniform regularization, the method assigns higher penalties to parameters that contribute most to the loss gap, which is used here as a proxy for privacy leakage. The authors support their approach with extensive experiments, comparing against defense techniques under various state-of-the-art membership inference attacks.

Strengths:
1. The main intuition for the defense method is well-motivated: parameters that are more sensitive to changes in the training data should be more "protected".
2. The method operates as a fine-tuning procedure, which makes it modular and compatible with other defense techniques.
3. The experimental results demonstrate consistent improvements over baseline defenses.

Weaknesses:
1. The dataset is not fully clear to me. For the PAST fine-tuning step, is the training set used as the member examples and the inference set as the non-member examples?
2. The training-plus-fine-tuning pipeline now depends not only on the training set but also on the data used as non-members during regularization. This raises the concern whether the model could leak information about the non-member set as well.

**Audience:**

Yes

**Audience Explanation:**

This paper could be of interest to researchers working on privacy attacks in machine learning and more broadly to the trustworthy machine learning community.

**Broader Impact Concerns:**

No concerns

**Claims And Evidence:**

Yes

**Claims Explanation:**

The paper is generally well-written, building up the motivation, the techniques and the experimental results in a  clear step-by-step way. The authors include several targeted experiments that address natural questions about the behavior of their defense technique.

**Requested Changes:**

Requested changes:
- Many citations are not in parenthetical form but should be.
- The symbol $m$ is overloaded. It denotes both the indicator of whether a sample is in the training dataset and the number of member examples in the definition of the loss gap. One of these could be assigned a different symbol to avoid ambiguity.

Recommended changes:
- The term "privacy sensitivity" may cause confusion to readers with a theoretical background in privacy. There, sensitivity typically refers to how much the model parameters change across different training datasets. In this paper,  it instead refers to how much the loss gap changes when a parameter changes. While the two notions are related from a privacy risk perspective, the terminology could be confusing.
- I would be interested in reading more about the limitations of this defense approach.

---

### Review · Reviewer_ANpf · 2026-05-31

**Summary Of Contributions:**

Dear authors, I'm sorry for the late review.

The authors introduce PAST, a new method to defend against Membership Inference attacks. The main idea is that not all the model parameters contribute in the same way to the privacy leakage. Therefore, some of them may need more regularization than others. The claim of the authors is that this PAST improves resistance to privacy attacks while also preserving model utility. PAST is applied as a post-training method, which makes it compatible with already existing defense methods.

The quantification of the privacy sensitivity is done by measuring the gradient of the loss gap wrt the parameters. The authors discovered that in the Cifar training with Resnet, only a small part of the parameters can have a high sensitivity. This can be used to promote sparsity for these parameters. The method is tested on different models and datasets.

**Audience:**

Yes

**Audience Explanation:**

I believe that the topic is interesting and would be of interest to the TMLR audience once authors solve the problems highlighted in the review

**Broader Impact Concerns:**

I do not have any ethical concerns about this work.

**Claims And Evidence:**

No

**Claims Explanation:**

- I believe that PAST has many of the problems described in the paper "Evaluations of Machine Learning Privacy Defenses are Misleading" [3]. In particular, the idea of minimizing a population-level privacy leakage is exactly what they criticized in [3] since this could hide the vulnerability of some specific samples. Moreover, there is no canary-based evaluation to target these specific outliers that could be present in the dataset. I would suggest authors to discuss this in their paper, reporting the metrics on the canary before and after the application of PAST. [3] also includes in its evaluation RelaxLoss, one of the methods also considered in this paper.
- In "Experiments Details", the authors stated that the models are trained with a certain set of hyperparameters. It would be helpful for reproducibility and also to evaluate the different results to understand how these parameters were chosen and if these were always the same for all the different datasets/models trained.
- In the experimental section, the authors did not mention either Differential Privacy nor the DP-SGD algorithm [1]. I believe that it is extremely important to compare PAST also with this baseline to understand how it compares both in terms of privacy leakage and utility with a formal method that allows for a formal guarantee of privacy. For instance, Opacus [2] can be used for this comparison since it includes state-of-the-art techniques to offer tradeoffs between epsilon guarantee and utility [4]. The hyperparameters in this case should be tuned accordingly to have a fair comparison with the other methods presented in the papers.
- Experiments are reported without std. Moreover, in a few of them, for instance Figure 5a, the differences between PAST vs w/o PAST are not too evident, a statistical significance test is not reported, but it would help to understand if PAST is actually helping in improving the P1 Score.
- In some experiments, \alpha=1.5 and in others \alpha=2.5, could authors explain these choices? In Table 1 in the appendix, 2.5 is never reported in the range of \alpha.
- Since PAST is a method that can be applied in a post-training phase, the authors also compared this solution with other defenses. In FIgure 5B, they showed the comparison of the P1 Score. The authors stated that PAST was used to fine-tune the methods trained with other approaches for an additional 50 epochs. It is not clear to me if in the comparison, the other methods were also trained for 150 epochs or not. If not, the comparison is not fair since the differences in terms of P1 Score could also be caused by the fact that we train the model for an additional 50 epochs.

Other minor things:
- In the introduction RMIA is mentioned without citation ("including state-of-the-art RMIA attack")


[1] Deep Learning with Differential Privacy https://arxiv.org/abs/1607.00133

[2] Opacus https://opacus.ai/

[3] Evaluations of Machine Learning Privacy Defenses are Misleading https://dl.acm.org/doi/abs/10.1145/3658644.3690194

[4] Unlocking High-Accuracy Differentially Private Image Classification through Scale https://arxiv.org/abs/2204.13650

**Requested Changes:**

I'd like authors to address the comments reported in the previous section